# Improving Adaptivity via Over-Parameterization in Sequence Models

**Yicheng Li**
Department of Statistics and Data Science
Tsinghua University, Beijing, China
`liyc22@mails.tsinghua.edu.cn`

**Qian Lin** *
Department of Statistics and Data Science
Tsinghua University, Beijing, China
`qianlin@tsinghua.edu.cn`

## Abstract

It is well known that eigenfunctions of a kernel play a crucial role in kernel regression. Through several examples, we demonstrate that even with the same set of eigenfunctions, the order of these functions significantly impacts regression outcomes. Simplifying the model by diagonalizing the kernel, we introduce an over-parameterized gradient descent in the realm of sequence model to capture the effects of various orders of a fixed set of eigen-functions. This method is designed to explore the impact of varying eigenfunction orders. Our theoretical results show that the over-parameterization gradient flow can adapt to the underlying structure of the signal and significantly outperform the vanilla gradient flow method. Moreover, we also demonstrate that deeper over-parameterization can further enhance the generalization capability of the model. These results not only provide a new perspective on the benefits of over-parameterization and but also offer insights into the adaptivity and generalization potential of neural networks beyond the kernel regime.

## 1 Introduction

In recent years, the remarkable success of neural networks in a wide array of machine learning applications has spurred a search for theoretical frameworks capable of explaining their efficacy and efficiency. One such framework is the Neural Tangent Kernel (NTK) theory (see, e.g., Jacot et al. [2018], Allen-Zhu et al. [2019]), which has emerged as a pivotal tool for understanding the dynamics of neural network training in the infinite-width limit. The NTK theory posits that the training dynamics of wide neural networks can be closely approximated by a kernel gradient descent method with the corresponding NTK, elucidating their convergence behaviors during gradient descent and shedding light on their generalization capabilities. Parallel to this, an extensive literature on kernel regression (see, e.g., Bauer et al. [2007], Yao et al. [2007]) has studied its generalization properties, showing its minimax optimality under certain conditions and providing insights into the bias-variance trade-off. Thus, one can almost fully understand the generalization properties of neural networks in the NTK regime by analyzing the kernel regression method.

However, the application of NTK theory to analyze neural networks, while invaluable, essentially frames the problem within a traditional statistical method by a fixed kernel. The NTK analysis, by its reliance on the fixed kernel approximation, can not entirely account for the adaptability and flexibility exhibited by neural networks, particularly those of finite width that deviate from the theoretical infinite-width limit [Woodworth et al., 2020]. Moreover, empirical evidence [Wenger et al., 2023, Seleznova and Kutyniok, 2022] also suggests that the assumption of a constant kernel during training, a cornerstone of NTK analysis, may not hold in practical scenarios where the network architecture or

---

*Corresponding author Qian Lin also affiliates with Beijing Academy of Artificial Intelligence, Beijing, China

38th Conference on Neural Information Processing Systems (NeurIPS 2024).

initialization conditions foster a dynamic evolution of the kernel. Also, Gatmiry et al. [2021] showed the benefits brought by the adaptivity of the kernel on a three-layer neural network. These results underscore the need for a more nuanced understanding of neural network training dynamics, one that considers the intricate interplay between network architecture, initialization, and the optimization process beyond the simplifications of NTK theory.

Recently, another branch of research has focused on the over-parameterization nature of neural networks beyond the NTK regime, exploring how over-parameterization can lead to implicit regularization and even improve the generalization. In terms of training dynamics, studies (Hoff [2017], Gunasekar et al. [2017], Arora et al. [2019a], Kolb et al. [2023], etc.) in this domain have revealed that over-parameterized models, particularly those trained with gradient descent and its variants, exhibit biases towards simpler, more generalizable functions, even in the absence of explicit regularization terms. Moreover, in terms of generalization, recent works [Vaškevičius et al., 2019, Zhao et al., 2022, Li et al., 2021a] have shown that in the setting of high dimensional linear regression, over-parameterized models with proper initialization and early stopping can achieve minimax optimal recovery under certain conditions. These results underscore the potential and benefits of over-parameterized models that go beyond the traditional statistical paradigms.

In this work, we will incorporate the insights from the kernel regression and the over-parameterization theory to investigate how over-parameterization can improve generalization and also adaptivity under the non-parametric regression framework. As a first step towards this direction, we will focus on the sequence model, which is an approximation of a wide spectrum of non-parametric models including kernel regression. We will show that, by dynamically adapting to the underlying structure of the signal during the training process, over-parameterization method with gradient descent can significantly improve the generalization properties compared with the fixed-eigenvalues method. We believe that our results provide a new perspective on the benefits of over-parameterization and offer insights into the adaptivity and generalization properties of neural networks beyond the NTK regime.

## 1.1 Our contributions

**Limitations of the (fixed) kernel regression.**   In this work, we first investigate the limitations of the (fixed) kernel regression method by specific examples, illustrating that the traditional kernel regression method suffers from the misalignment between the kernel and the truth function. We show that even when the eigen-basis of the kernel is fixed, the associated eigenvalues, particularly their alignment with the truth function's coefficients in the eigen-basis, can significantly affect the generalization properties of the method.

**Advantages of over-parameterized gradient descent.**   Focusing on the alignment between the kernel's eigenvalues and the truth signal (the truth function's coefficients), we consider the sequence model and introduce an over-parameterization method (8) that can dynamically adjust the eigenvalues during the learning process. We show that with proper early-stopping, the over-parameterization method can achieve nearly the oracle convergence rate regardless of the underlying structure of the signal, significantly outperforming the vanilla fixed-eigenvalues method when the misalignment is severe. In addition, the over-parameterization method is also adaptive by its universal choice of the stopping time, which is independent of the signal's structure.

**Benefits of deeper parameterization.**   Moreover, we also consider deeper over-parameterization (14) and explore how depth affects the generalization properties of the over-parameterization method. Our results show that adding depth can further ease the impact of the initial choice of the eigenvalues, thus improving the generalization capability of the model. We also provide numerical experiments to validate our theoretical results in Section C.

## 1.2 Notations

We denote by $\ell^2 = \left\{ (a_j)_{j \geq 1} \mid \sum_{j \geq 1} a_j^2 < \infty \right\}$ the space of square summable sequences. We write $a \lesssim b$ if there exists a constant $C > 0$ such that $a \leq Cb$ and $a \asymp b$ if $a \lesssim b$ and $b \lesssim a$, where the dependence of the constant $C$ on other parameters is determined by the context.

## 2 Limitations of Fixed Kernel Regression

Let us consider the non-parametric regression problem given by $y = f^*(x) + \varepsilon$, where $\varepsilon$ is the noise with mean zero and variance $\sigma^2$, $x \in \mathcal{X}$ and $\mathcal{X}$ is the input space with $\mu$ being a probability measure supported on $\mathcal{X}$. The function $f^*(x)$ represents the unknown regression function we aim to learn. Suppose we are given samples $\{(x_i, y_i)\}_{i=1}^n$, drawn i.i.d. from the model. We denote $X = (x_1, \ldots, x_n)^\top$ and $Y = (y_1, \ldots, y_n)^\top$.

Let $k : \mathcal{X} \times \mathcal{X} \to \mathbb{R}$ be a continuous positive definite kernel and $\mathcal{H}_k$ be its associated reproducing kernel Hilbert space (RKHS). The well-known Mercer's decomposition [Steinwart and Scovel, 2012] of the kernel function $k$ gives

$$k(x, y) = \sum_{j=1}^{\infty} \lambda_j e_j(x) e_j(y), \tag{1}$$

where $(e_j)_{j \geq 1}$ is an orthonormal basis of $L^2(\mathcal{X}, \mathrm{d}\mu)$, and $(\lambda_j)_{j \geq 1}$ are the eigenvalues of $k$ in descending order. Moreover, we can introduce the feature map $\Phi(x) = (\lambda_j^{\frac{1}{2}} e_j(x))_{j \geq 1} : \mathcal{X} \to \ell^2$ (as a column vector) such that $k(x, x') = \langle \Phi(x), \Phi(x') \rangle$. With the feature map, a function $f \in \mathcal{H}_k$ can be represented as $f(x) = \langle \Phi(x), \beta \rangle_{\ell^2}$ for some $\beta \in \ell^2$.

Defining the empirical loss as $L = \frac{1}{2n} \sum_{i=1}^n (y_i - f(x_i))^2$, we can consider an estimator $f_t = \langle \Phi(x), \beta_t \rangle_{\ell^2}$ governed by the following gradient flow on the feature space

$$\dot{\beta}_t = -\nabla_\beta L = \frac{1}{n} \sum_{i=1}^n (y_i - \langle \Phi(x_i), \beta_t \rangle_{\ell^2}) \Phi(x_i), \quad \text{where} \quad \beta_0 = \mathbf{0}. \tag{2}$$

This kernel gradient descent (flow) estimator corresponds to neural networks at infinite width limit by the celebrated neural tangent kernel (NTK) theory [Jacot et al., 2018, Allen-Zhu et al., 2019].

An extensive literature [Yao et al., 2007, Lin et al., 2018, Li et al., 2024a] has studied the generalization performance of such kernel gradient descent estimator. From the Mercer's decomposition, we can further introduce interpolation spaces for $s \geq 0$ as

$$[\mathcal{H}_k]^s := \Big\{ \sum_{j=1}^{\infty} \beta_j \lambda_j^{\frac{s}{2}} e_j \mid (\beta_j)_{j \geq 1} \in \ell^2 \Big\}, \tag{3}$$

which is equipped with the norm $\|f\|_{[\mathcal{H}_k]^s} = \|\boldsymbol{\beta}\|_{\ell^2}$ for $f = \sum_{j=1}^{\infty} \beta_j \lambda_j^{\frac{s}{2}} e_j$. Particularly, the interpolation space $[\mathcal{H}_k]^1$ corresponds to the RKHS $\mathcal{H}_k$ itself. Then, assuming the eigenvalue decay rate $\lambda_j \asymp j^{-\gamma}$, the standard results (see, e.g., Yao et al. [2007], Li et al. [2024a]) in kernel regression state that the optimal rate of convergence under the source condition $f^* \in [\mathcal{H}_k]^s$ with $\|f^*\|_{[\mathcal{H}_k]^s} \leq 1$ is $n^{-\frac{s\gamma}{s\gamma+1}}$. However, since the interpolation space $[\mathcal{H}_k]^s$ is defined via the eigen-decomposition of the kernel, the generalization performance of kernel regression methods is ultimately related to the eigen-decomposition of the kernel and the decomposition of the target function under the basis, so the performance is intrinsically limited by the relation between the target function and the kernel itself. In other words, the choice of the kernel could affect the performance of the method. To demonstrate this quantitatively, let us consider the following examples.

**Example 2.1** (Eigenfunctions in common order). It is well known that kernels possessing certain symmetries, such as dot-product kernels on the sphere or translation-invariant periodic kernels on the torus, share the same set of eigenfunctions (such as the spherical harmonics or the Fourier basis). If we consider a fixed set of eigenfunctions $\{e_j\}_{j \geq 1}$ and a given truth function $f^*$, for two kernels $k_1$ and $k_2$ with eigenvalue decay rates $\lambda_{j,1} \asymp j^{-\gamma_1}$ and $\lambda_{j,2} \asymp j^{-\gamma_2}$ respectively, it follows that:

$$f^* \in [\mathcal{H}_{k_1}]^{s_1} \iff f^* \in [\mathcal{H}_{k_2}]^{s_2} \quad \text{for} \quad \gamma_1 s_1 = \gamma_2 s_2.$$

Given that the convergence rate is dependent solely on the product $s\gamma$, the convergence rates relative to the two kernels will be identical.

Example 2.1 seems to show that when the eigenfunctions are fixed, kernel regression methods yield similar performance across different kernels. However, it's important to note that this similarity is due

to both kernels having *the same eigenvalue decay order*, which aligns with the predetermined order of the basis. In fact, if the eigenvalue decay order of a kernel deviates from that of the true function, even if the eigenfunction basis remain the same, it can lead to significantly different convergence rates. Let us consider the following example to illustrate this point.

**Example 2.2** (Low-dimensional structure). Consider translation-invariant periodic kernels on the torus $\mathbb{T}^d = [-1,1)^d$ with the uniform distribution. Then, their eigenfunctions are given by the Fourier basis $\phi_{\boldsymbol{m}}(x) = \exp(i\pi \langle \boldsymbol{m}, x \rangle)$, $\boldsymbol{m} \in \mathbb{Z}^d$. Within this basis, a target function $f^*(x)$ can be represented as:

$$f^* = \sum_{\boldsymbol{m} \in \mathbb{Z}^d} f_{\boldsymbol{m}} \phi_{\boldsymbol{m}}(x).$$

Assuming $f^*$ exhibits a low-dimensional structure, specifically $f^*(x) = g(x_1, \ldots, x_{d_0})$ for some $d_0 < d$, and considering $g$ belongs to the Sobolev space $H^t(\mathbb{T}^{d_0})$ of order $t$, the coefficients $f_{\boldsymbol{m}}$ can be shown to simplify to:

$$f_{\boldsymbol{m}} = \begin{cases} g_{\boldsymbol{m}_1}, & \boldsymbol{m} = (\boldsymbol{m}_1, \boldsymbol{0}), \ \boldsymbol{m}_1 \in \mathbb{Z}^{d_0}, \\ 0, & \text{otherwise.} \end{cases}$$

Let us now consider two translation-invariant periodic kernels $k_1$ and $k_2$ given in terms of their eigenvalues: $k_1$ is given by $\lambda_{\boldsymbol{m},1} = (1 + \|\boldsymbol{m}\|^2)^{-r}$ for some $r > d/2$, whose RKHS is the full-dimensional Sobolev space $H^r(\mathbb{T}^d)$; $k_2$ is given by $\lambda_{\boldsymbol{m},2} = (1 + \|\boldsymbol{m}\|^2)^{-r}$ for $\boldsymbol{m} = (\boldsymbol{m}_1, \boldsymbol{0})$ and $\lambda_{\boldsymbol{m},2} = 0$ otherwise. Then, the function $f^*$ belongs to both $[\mathcal{H}_{k_1}]^s$ and $[\mathcal{H}_{k_2}]^s$ for $s = t/r$. After reordering the eigenvalues in descending order, the decay rates for the two kernels are identified as $\gamma_1 = 2r/d$ and $\gamma_2 = 2r/d_0$. Thus, the convergence rates with respect to the two kernels are respectively:

$$\frac{2t}{2t + d} \quad \text{and} \quad \frac{2t}{2t + d_0}.$$

Therefore, we see that when $d$ is significantly larger than $d_0$, the convergence rate for the second kernel notably surpasses that of the first.

This example illustrates that the eigenvalues can significantly impact the learning rate, even when the eigenfunctions are the same. In the scenario presented, the second kernel benefits from the low-dimensional structure of the target function by focusing only on the relevant dimensions, whereas the first one suffers from the curse of dimensionality since it considers all dimensions. The key point to take away from this example is the *alignment between the kernel and the target function*. To generalize this example, we can consider the following example where the order of the eigenvalues does not align with the order of the target function's coefficients.

**Example 2.3** (Misalignment). Let us fix a set of the eigenfunctions $(e_j)_{j\geq 1}$ and expand the truth function as $f^* = \sum_{j\geq 1} \theta_j^* e_j$. Note that by giving $(e_j)_{j\geq 1}$, we already defined an order of the basis in $j$, but coefficients $\theta_j^*$ of the truth function are not necessarily ordered by $j$. Suppose that an index sequence $\ell(j)$ gives the descending order of $\left|\theta_{\ell(j)}^*\right|$. Then we can characterize the misalignment by the difference between $\ell(j)$ and $j$. Specifically, we assume that

$$\left|\theta_{\ell(j)}^*\right| \asymp j^{-(p+1)/2} \quad \text{and} \quad \ell(j) \asymp j^q \quad \text{for} \quad p > 0, \ q \geq 1, \tag{4}$$

where larger $q$ indicates a more severe misalignment. In terms of eigenvalues, let us consider $\lambda_{j,1} \asymp j^{-\gamma}$, which is in the order of $j$, while $\lambda_{\ell(j),2} \asymp j^{-\gamma}$, which is in the order of $\ell(j)$. Then, the convergence rates with the two sequences of coefficients are respectively

$$\frac{p}{p+q} \quad \text{and} \quad \frac{p}{p+1}.$$

Therefore, the convergence rates can differ greatly if the misalignment is significant, namely when $q$ is large.

From Example 2.2 and Example 2.3, we find that it is beneficial that *the eigenvalues of the kernel align with the structure of the target function*. However, one can hardly choose the proper kernel a priori, especially when the structure of the target function is unknown, so the fixed kernel regression can be limited by the kernel itself and be unsatisfactory. Motivated by these examples, we would like to explore the idea of an "adaptive kernel approach," where the kernel can be learned from the data.

# 3 Adapting the Eigenvalues by Over-parameterization in the Sequence Model

Motivated by the examples in the last section, as a first step toward the adaptive kernel approach, we consider *adapting the eigenvalues of the kernel with eigenfunctions fixed*. To simplify the analysis, we would like to the following sequence model, which captures the essences of many statistical models [Brown et al., 2002, Johnstone, 2017].

**The sequence model**   Let us consider the sequence model [Johnstone, 2017]

$$z_j = \theta_j^* + \xi_j, \quad j \geq 1 \tag{5}$$

where $(z_j)_{j \geq 1}$ is the observation, $\boldsymbol{\theta}^* = (\theta_j^*)_{j \geq 1} \in \ell^2$ is a sequence of unknown truth parameters and $\xi_j$, $j \geq 1$ are (not necessarily independent) $\epsilon^2$-sub-Gaussian random variables with mean zero and variance at most $\epsilon^2$. For any estimator $\hat{\boldsymbol{\theta}} = (\hat{\theta}_j)_{j \geq 1}$, the generalization error is measured by $\mathcal{R}(\hat{\boldsymbol{\theta}}; \boldsymbol{\theta}^*) = \sum_{j=1}^{\infty} (\hat{\theta}_j - \theta_j^*)^2$. Under the asymptotic framework, we are often interested in the behavior of the generalization error as $\epsilon \to 0$. Here, we note that the connection between non-parametric regression and the sequence model yields $\epsilon^2 \asymp n^{-1}$.

To see the connection between the sequence model and the non-parametric regression model, we first write the gradient flow (2) in the RKHS in the matrix form as

$$\dot{\beta}_t = -\nabla_\beta \mathcal{L} = -\frac{1}{n} \Phi(X) \Phi(X)^\top \beta_t + \frac{1}{n} \Phi(X) \boldsymbol{y},$$

where the feature matrix $\Phi(X) = (\Phi(x_1), \ldots, \Phi(x_n))_{\infty \times n}$ and $\boldsymbol{y} = (y_1, \ldots, y_n)^\top$. Now, since the eigenfunctions $(e_j)_{j \geq 1}$ are fixed, intuitively, the gradient flow can be diagonalized in the eigen-basis since $\frac{1}{n} \Phi(X) \Phi(X)^\top \approx \Lambda = \mathrm{diag}(\lambda_1, \lambda_2, \ldots)$ and the noise components are approximately normal with variance $\sigma^2/n$ by the central limit theorem. Thus, we reach the sequence model. We refer to Subsection B.1 for a more detailed explanation of the connection between the sequence model and the kernel regression model.

Regarding the power series expansion (3) in RKHS, for a sequence $(\lambda_j)_{j \geq 1}$ of descending positive numbers (e.g., $\lambda_j = j^{-\gamma}$), we can consider similarly the parameterization $\theta_j = \lambda_j^{\frac{1}{2}} \beta_j$, $j \geq 1$ in $\ell^2$. Since $(\lambda_j)_{j \geq 1}$ corresponds to the eigenvalues of the kernel in the kernel regression, here we also refer to $(\lambda_j)_{j \geq 1}$ as the *"eigenvalues"* with a little abuse of terminology.

With the component-wise loss function $L_j(\theta_j) = \frac{1}{2}(\theta_j - z_j)^2$, we can apply a gradient descent (gradient flow) with early stopping to derive a component-wise estimator $\hat{\theta}_j$. If we directly parameterize $\theta_j = \lambda_j^{\frac{1}{2}} \beta_j$ with only $\beta_j$ trainable, we obtain the vanilla gradient descent method, which is just the diagonalized version of the kernel gradient descent. The estimator is simply given by $\hat{\theta}_j = (1 - e^{-\lambda_j t}) z_j$, where $t$ is the stopping time, and its generalization error is easily computed as

$$\mathbb{E} \mathcal{R}(\hat{\boldsymbol{\theta}}^{\mathrm{GF}}; \boldsymbol{\theta}^*) = B_{\mathrm{GF}}^2(t; \boldsymbol{\theta}^*) + \epsilon^2 V_{\mathrm{GF}}(t) = \sum_{j=1}^{\infty} \left(e^{-\lambda_j t} \theta_j^*\right)^2 + \epsilon^2 \sum_{j=1}^{\infty} \left(1 - e^{-\lambda_j t}\right)^2. \tag{6}$$

We note here that these quantities also correspond to generalization error in the (fixed) kernel regression setting [Li et al., 2024a]. In particular, under the setting of (4) and $\lambda_j \asymp j^{-\gamma}$, by choosing $t \asymp \epsilon^{-\frac{2q\gamma}{p+q}}$, we obtain the convergence rate $\epsilon^{\frac{2p}{p+q}}$, which is far from optimal if $q$ is large.

## 3.1 Over-parameterized gradient descent

By the discussion in the previous section, we find it essential to adjust the eigenvalues beyond the fixed ones $(\lambda_j)_{j \geq 1}$. Inspired by the over-parameterization nature of neural networks, we can also consider over-parameterization with gradient descent in our sequence model to train the eigenvalues: Replacing $\lambda_j^{1/2}$ with trainable parameter $a_j$, let us parameterize

$$\theta_j = a_j \beta_j, \tag{7}$$

where $a_j$ aims to learn the eigenvalues and $\beta_j$ aims to learn the signal. We consider the following gradient flow (simultaneously for each component $j$):

$$\dot{a}_j = -\nabla_{a_j} L_j, \quad \dot{\beta}_j = -\nabla_{\beta_j} L_j,$$
$$a_j(0) = \lambda_j^{1/2}, \quad \beta_j(0) = 0. \tag{8}$$

Here, $(\lambda_j)_{j \geq 1}$ serves as the initial eigenvalues, while the trainable parameters $(a_j)_{j \geq 1}$ are updated to adjust the eigenvalues during the training process.

To state our results with the most generality, let us introduce the following quantities on the target parameter sequence $\boldsymbol{\theta}^*$:

$$J_{\text{sig}}(\delta) := \left\{ j : \left| \theta_j^* \right| \geq \delta \right\}, \quad \Phi(\delta) := |J_{\text{sig}}(\delta)|, \quad \Psi(\delta) = \sum_{j \notin J_{\text{sig}}(\delta)} (\theta_j^*)^2. \tag{9}$$

The quantity $\Phi(\delta)$ measures the number of significant components in the target parameter sequence $\boldsymbol{\theta}$, while $\Psi(\delta)$ measures the contribution of the insignificant components, which are commonly considered in the literature on the sequence model [Johnstone, 2017]. For the concrete setting of (4), it is easy to show that

$$\Phi(\delta) \asymp \delta^{-\frac{2}{p+1}}, \quad \Psi(\delta) \asymp \delta^{\frac{2p}{p+1}}. \tag{10}$$

Moreover, we make the following assumption on the span of the significant components.

**Assumption 1.** There exists constants $\kappa \geq 0$ and $C_{\text{sig}} > 0$ such that

$$\max J_{\text{sig}}(\delta) \leq C_{\text{sig}} \delta^{-\kappa}, \quad \forall \delta > 0. \tag{11}$$

Assumption 1 says that the span of the significant components, namely those with $\left| \theta_j^* \right| \geq \delta$, grows at most polynomially in $1/\delta$. This assumption is mild and holds for many practical settings, such as cases considered in Example 2.3 ($\kappa = \frac{2q}{p+1}$ for the first kernel). In other perspective, it imposes a mild condition on the misalignment between the ordering of the truth signal and the ordering of the eigenvalues, where $\kappa$ measures the misalignment between the ordering of $\theta_j$ and the ordering of $j$ itself. Then, the following theorem characterizes the generalization error of the resulting estimator.

**Theorem 3.1.** *Consider the sequence model* (5) *under Assumption 1. Fix* $\lambda_j \asymp j^{-\gamma}$ *for some* $\gamma > 1$ *and let* $\hat{\boldsymbol{\theta}}^{\text{Op}}$ *be the estimator given by the gradient flow* (8) *stopped at time $t$. Then, there exists some constants* $B_1, B_2 > 0$ *such that when* $B_1 \epsilon^{-1} \leq t \leq B_2 \epsilon^{-1}$*, we have*

$$\mathbb{E} \mathcal{R}(\hat{\boldsymbol{\theta}}^{\text{Op}}, \boldsymbol{\theta}^*) \lesssim \epsilon^2 \left[ \Phi(\epsilon) + \epsilon^{-1/\gamma} \right] + \Psi(\epsilon \ln(1/\epsilon)) \quad as \quad \epsilon \to 0. \tag{12}$$

### 3.2 Towards deeper over-parameterization

Let us further introduce deeper over-parameterization by adding extra $D$-layers:

$$\theta_j = a_j b_j^D \beta_j \tag{13}$$

and consider the gradient flow

$$\dot{a}_j = -\nabla_{a_j} L_j, \quad \dot{b}_j = -\nabla_{b_j} L_j, \quad \dot{\beta}_j = -\nabla_{\beta_j} L_j,$$
$$a_j(0) = \lambda_j^{1/2}, \quad b_j(0) = b_0 > 0, \quad \beta_j(0) = 0, \tag{14}$$

where $b_0$ is the common initialization of all $b_j$. We remark here if one considers the over-parameterization $\theta_j = a_j b_{j,1} \cdots b_{j,D} \beta_j$ with the same initialization $b_{j,k} = b_0$, $k = 1, \ldots, D$, then $b_{j,k}$'s remain to be the same by symmetry, so this is equivalent to our parameterization $\theta_j = a_j b_j^D \beta_j$. The following theorem presents an upper bound for the generalization error by this deeper over-parameterized gradient flow.

**Theorem 3.2.** *Consider the sequence model* (5) *under Assumption 1. Fix* $\lambda_j \asymp j^{-\gamma}$ *for some* $\gamma > 1$ *and let* $\hat{\boldsymbol{\theta}}^{\text{Op},D}$ *be the estimator given by the gradient flow* (14) *stopped at time $t$. Then, by choosing* $b_0 \asymp \epsilon^{\frac{1}{D+2}}$*, there exists some constants* $B_1, B_2 > 0$ *such that when* $B_1 \epsilon^{-1} \leq b_0^D t \leq B_2 \epsilon^{-1}$*, we have*

$$\mathbb{E} \mathcal{R}(\hat{\boldsymbol{\theta}}^{\text{Op},D}, \boldsymbol{\theta}^*) \lesssim \epsilon^2 \left[ \Phi(\epsilon) + \epsilon^{-\frac{2}{D+2} \frac{1}{\gamma}} \right] + \Psi(\epsilon \ln(1/\epsilon)) \quad as \quad \epsilon \to 0. \tag{15}$$

### 3.3 Discussion of the results

**Benefits of Over-parameterization**   Theorem 3.1 and Theorem 3.2 demonstrate the advantage of over-parameterization in the sequence model. Compared with the vanilla fixed-eigenvalues gradient descent method, the over-parameterized gradient descent method can significantly improve the generalization performance by adapting the eigenvalues to the truth signal. For a more concrete example, if we consider the setting of (4), plugging (10) yields the following corollary.

**Corollary 3.3.** *Consider the over-parameterized gradient descent in* (8) *(setting $D = 0$) or* (14). *Suppose* (4) *holds and $\lambda_j \asymp j^{-\gamma}$ for $\gamma > 1$ and $\gamma \geq \frac{p+1}{D+2}$. Then, by choosing $b_0 \asymp \epsilon^{\frac{1}{D+2}}$ (if $D \neq 0$) and $t \asymp \epsilon^{-\frac{2D+2}{D+2}}$, we have*

$$\mathbb{E}\mathcal{R}(\hat{\boldsymbol{\theta}}^{\mathrm{Op,D}}, \boldsymbol{\theta}^*) \lesssim \epsilon^{\frac{2p}{p+1}} (\ln(1/\epsilon))^{\frac{2p}{p+1}} \quad as \quad \epsilon \to 0. \tag{16}$$

*In comparison, the vanilla gradient flow method yields the rate $\epsilon^{\frac{2p}{p+q}}$.*

Ignoring the logarithmic factor, Corollary 3.3 shows that the over-parameterized gradient descent method can achieve a nearly optimal rate $\epsilon^{\frac{2p}{p+1}}$, while the vanilla gradient descent method only achieves the rate $\epsilon^{\frac{2p}{p+q}}$. The improvement is significant when $q$ is large, which corresponds to the case that the misalignment between the ordering of the truth signal and the ordering of the eigenvalues is severe. Moreover, if we return to the low-dimensional regression function in Example 2.2 with the isotropic kernel $k_1$, we can see that while the vanilla gradient descent method suffers from the curse of dimensionality with the rate $\frac{2t}{2t+d}$, the over-parameterization leads to the dimension-free rate $\frac{2t}{2t+d_0}$. Therefore, the over-parameterization significantly improves the generalization performance.

**Learning the eigenvalues**   To further investigate how the eigenvalues are adapted by over-parameterized gradient descent, we present the following proposition.

**Proposition 3.4.** *Given the same conditions as in Theorem 3.2 or Theorem 3.1 (with $D = 0$ and $b_j^D = 1$ for Theorem 3.1), the term learning the eigenvalues $a_j(t)b_j^D(t)$ is non-decreasing in $t$ for each $j$. Moreover, letting $\delta \in (0,1)$, when $\epsilon$ is small enough, the following holds at time $t$ chosen as in Theorem 3.1 or Theorem 3.2:*

- *Signal component: There exist constants $C, c > 0$ such that for any component satisfying $|\theta_j^*| \geq C\epsilon \ln(1/\epsilon)$, it holds with probability at least $1 - \delta$ that*

$$a_j(t)b_j^D(t) \geq c|\theta_j^*|^{\frac{D+1}{D+2}}. \tag{17}$$

- *Noise component: There exist constants $c, C, C' > 0$ such that, for any component where $|\theta_j^*| \leq \epsilon$ and $\lambda_j \leq c\epsilon^{\frac{2}{D+2}}$, it holds with probability at least $1 - \delta$ that*

$$a_j(t)b_j^D(t) \leq C\lambda_j^{\frac{1}{2}}\epsilon^{\frac{D}{D+2}} = C'a_j(0)b_j^D(0). \tag{18}$$

From this proposition, we can see that for the signal components, the eigenvalues are learned to be at least a constant times a certain power of the truth signal magnitude. Thus, over-parameterized gradient descent adjusts the eigenvalues to match the truth signal as expected. In the case of noise components, although the eigenvalues are still increasing due to the training process, the eigenvalues do not exceed the initial values by some constant factor, provided that $\lambda_j$ is relatively small. This finding suggests that over-parameterized gradient descent effectively adapts eigenvalues to the truth signal while mitigating overfitting to noise. We remark that when $\lambda_j$ is relatively large, the method still tends to overfit the noise components, contributing an extra $\epsilon^{-\frac{2}{D+2}\frac{1}{\gamma}}$ term in the generalization error, but this term becomes negligible for large $\gamma$. Moreover, we also remark that there is a $\ln(1/\epsilon)$ gap between the signal and noise components. This is because the signal and the noise can not be distinguished for the components in the middle.

**Adaptive choice of the stopping time**   A notable advantage of the over-parameterized gradient descent method is its adaptivity. Consider the scenario described by (4), vanilla gradient descent requires the selection of a stopping time $t \asymp \epsilon^{-(2q\gamma)/(p+q)}$ to achieve the optimal convergence

rate. However, this choice of stopping time critically depends on the unknown parameters $p$ and $q$ of the truth parameter, posing a significant challenge in practical applications. In contrast, the over-parameterized gradient descent only need to choose the stopping time as $t \asymp \epsilon^{-\frac{2D+2}{D+2}}$, which does not rely on the unknown truth parameters, while still achieving the nearly optimal convergence rate. This independence from the truth parameters allows the over-parameterization approach to adaptively accommodate any truth parameter structure by employing a fixed stopping time, without the need for prior knowledge about the truth function's properties.

**Effect of the depth** The results in Theorem 3.2 also show that deeper over-parameterization can further improve the generalization performance. In the two-layer over-parameterization, the extra term $\epsilon^{-1/\gamma}$ in Theorem 3.1 emerges due to the limitation of the adapting large eigenvalues. With the introduction of depth, namely adding extra $D$ layers to the model with proper initialization, this term can be improved to $\epsilon^{-\frac{2}{D+2}\frac{1}{\gamma}}$ in Theorem 3.2. This improvement suggests that the depth can refine the model's sensitivity to eigenvalue adaptation, enabling a more nuanced adjustment to the underlying signal structure. This finding underscores the importance of model depth in enhancing the learning process, providing also theoretical evidence for the empirical success of deep learning models.

**Comparison with previous works** We will compare our results with the existing literature [Zhao et al., 2022, Li et al., 2021a, Vaškevičius et al., 2019] on the generalization performance of over-parameterized gradient descent in the following aspects:

- **Problem settings:** While the existing literature [Zhao et al., 2022, Li et al., 2021a, Vaškevičius et al., 2019] investigate the realms of high-dimensional linear regression, focusing on implicit regularization and sparsity, the present study dives into kernel regression and its approximation by Gaussian sequence models, emphasizing the adaptivity of over-parameterization to the underlying signal's structure, a leap towards understanding model complexity beyond mere regularization. Moreover, while the literature primarily focuses on the setting of strong signal, weak signal and noise separation, we consider the more general setting of the sequence model with arbitrary signal and noise components.

- **Over-parameterization setup:** The existing work Zhao et al. [2022] considers the over-parameterization setup by the two-layer Hadamard product $\theta = a \odot b$ where the initialization is the same for each component that $a(0) = \alpha\mathbf{1}$ and $b(0) = \mathbf{0}$. In comparison, our work considers initializing the eigenvalues $a_j(0) = \lambda_j^{1/2}$ differently for each component. Moreover, we extend the over-parameterization to deeper models by adding extra $D$ layers. Although Vaškevičius et al. [2019] and the subsequent work Li et al. [2021a] also consider the deeper over-parameterization, their over-parameterization is in the form of $\theta = u^{\odot D} - v^{\odot D}$ with $u(0) = v(0) = \alpha\mathbf{1}$. Unfortunately, though being easy to analysis because of the homogeneous initialization, this setup could not bring insights into the learning of the eigenvalues, which is the key to our results. Furthermore, the analysis for Theorem 3.2 involves the interplay between the differently initialized $a_j$ and $b_j$, so our analysis is more involved than the existing works. We also remark that although we only consider the gradient flow in the analysis, the results can be extended to the gradient descent with proper learning rates.

- **Interpretation of the over-parameterization:** The previous works view the over-parameterization mainly as a mechanism for implicit regularization, while our work provides a novel perspective that over-parameterization adapts to the structure of the truth signal by learning the eigenvalues. Our theory also aligns with the neural network literature [Yang and Hu, 2022, Ba et al., 2022], where over-parameterization with gradient descent is known to be beneficial in learning the structure of the target function.

- **Connection to sparse recovery:** Our results can be phrased for the setting of high dimensional regression with sparsity. Taking a sparse signal $(\theta_j^*)_{j \geq 1}$, e.g., $\theta_j^* = 1$ for $j \in S$, $|S| = s$ and $\theta_j^* = 0$ for $j \notin S$, we find that $\Phi(\epsilon) = s$ and $\Psi(\epsilon) = 0$. Consequently, ignoring the extra error term, the resulting rate obtained by Theorem 3.1 or Theorem 3.2 is $\tilde{O}(s/n)$ (ignoring the logarithmic factor). This rate coincides with the minimax rate for sparse recovery in high-dimensional regression.

### 3.4 Proof outline

In this subsection, we will provide an outline of the proof of Theorem 3.1 and Theorem 3.2. For the detailed proof, we refer to Section D for the analysis of the gradient flow equation and Section E for the generalization error.

**Equation analysis**    The proof of Theorem 3.1 and Theorem 3.2 relies on the analysis of the gradient flow (8) and (14) for each component $j$. For notation simplicity, we will suppress the index $j$ in the following discussion. Firstly, the symmetry of the equation allows us to consider only the case $z > 0$. Then, one can find that

$$\frac{\mathrm{d}}{\mathrm{d}t}a^2 = \frac{\mathrm{d}}{\mathrm{d}t}\beta^2 = \frac{1}{D}\frac{\mathrm{d}}{\mathrm{d}t}b^2 = 2\theta(z-\theta),$$

so we get the conservation quantities $a^2(t) \equiv \beta^2(t) + \lambda$ and $b^2(t) \equiv D\beta^2(t) + b_0^2$.

Consequently, for the case $D = 0$, we can obtain the explicit gradient flow of $\theta$:

$$\dot{\theta} = \sqrt{a_0^4 + 4\theta^2}(z-\theta), \quad \theta(0) = 0.$$

Since $\sqrt{a_0^4 + 4\theta^2}$ can be bounded by a multiple of $a_0^2 + 2\theta$, we can consider the other equation $\dot{\theta} = (a_0^2 + 2\theta)(z - \theta)$, which admits a closed-form solution.

For the case $D \geq 1$, the equation is more complicated. We will apply a multiple stage analysis concerning both the effect of $a(t)$ and $b(t)$.

**The generalization error**    In terms of the generalization error, we first separate the noise case when $|\xi_j| \geq |\theta_j^*|/2$ and the signal case when $|\xi_j| < |\theta_j^*|/2$. For the noise case, we apply the analysis of the equation to show that $\theta_j(t)$ is bounded roughly by $\lambda_j$ for our choice of $t$. Moreover, the fact that $\lambda_j$ is summable ensures that error of these noise components does not sum up to infinity. On the other hand, for the signal case, if $|\theta_j^*| \geq \epsilon \ln(1/\epsilon)$, we can show that our choice of $t$ allows $\theta_j(t)$ to exceed $z/2$ and converge to $z$ close enough, so the error in these components is only caused by the random noise and sum up to $\epsilon^2 \Phi(\epsilon)$. In addition, the remaining signal components contribute to the error term $\Psi(\epsilon \ln(1/\epsilon))$. Summing up these two terms, we can obtain the desired generalization error bound.

## 4 Numerical Experiments

In this section, we provide some numerical experiments to validate the theoretical results. For more detailed numerical experiments, please refer to Section C.

We approximate the gradient flow equation (22) and (30) by discrete-time gradient descent and truncate the sequence model to the first $N$ terms for some very large $N$. We consider the settings as in Corollary 3.3 that $\boldsymbol{\theta}^*$ is given by (4) for some $p > 0$ and $q \geq 1$. We set $\epsilon^2 = n^{-1}$, where $n$ can be regarded as the sample size, and consider the asymptotic generalization error rates as $n$ grows.

We first compare the generalization error rates between vanilla gradient descent and over-parameterized gradient descent (OpGD) in Figure 1 on page 10. The results show that the over-parameterized gradient descent can achieve the optimal generalization error rate, while the vanilla gradient descent suffers from the misalignment caused by $q > 1$ and thus has a slower convergence rate. Moreover, with a logarithmic least-squares fitting, we find that the resulting generalization error rates also match the theoretical results in Corollary 3.3 (0.5 for OpGD and 0.33 for vanilla GD).

Additionally, we investigate the evolution of the eigenvalue terms $a_j(t)b_j^D(t)$ over time $t$ as discussed in Proposition 3.4. The results are shown in Figure 2 on page 10. We find that the eigenvalue terms can indeed adapt to the underlying structure of the signal: for large signals, the eigenvalue terms approach the signals as the training progresses, while for small signals, the eigenvalue terms do not increase significantly. Moreover, we find that deeper over-parameterization reduces the fluctuations of the eigenvalue terms for the noise components, and thus improves the generalization performance of the model.

In summary, the numerical experiments validate our theoretical results and provide insights into the adaptivity and generalization properties of the over-parameterized gradient descent method.

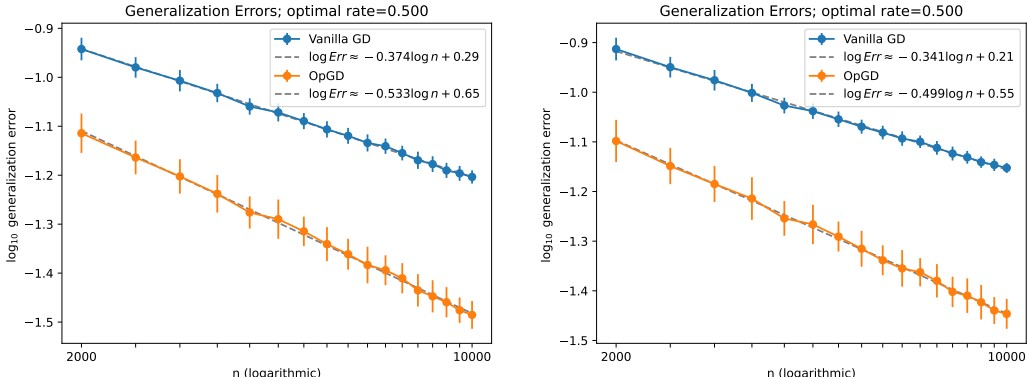

Figure 1: Comparison of the generalization error rates between vanilla gradient descent and over-parameterized gradient descent (OpGD). We set $p = 1$ and $q = 2$ for the truth parameter $\boldsymbol{\theta}^*$, and $\gamma = 1.5$ for the left column and $\gamma = 3$ for the right column. For each $n$, we repeat $64$ times and plot the mean and the standard deviation.

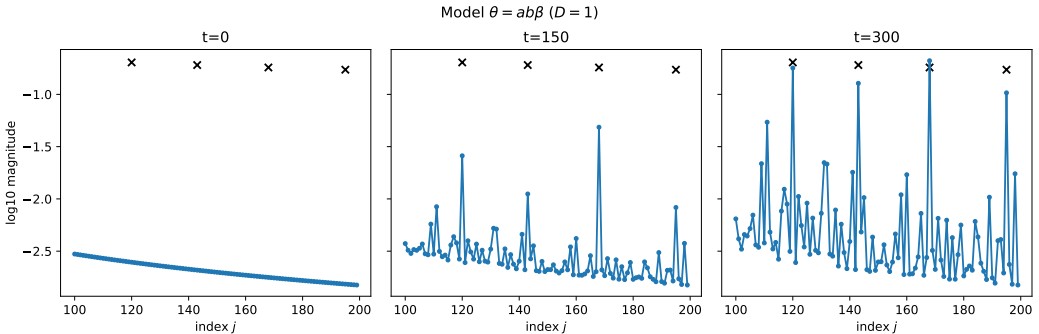

Figure 2: The evolution of the trainable eigenvalues $a_j(t)b_j^D(t)$ over the time $t$ across components $j = 100$ to $200$ for $D = 1$. The blue line shows the eigenvalues and the black marks show the non-zero signals scaled according to Proposition 3.4. For the settings, we set $p = 1$, $q = 2$ and $\gamma = 2$.

## 5 Conclusion and Future Work

In this work, we studied the generalization properties of over-parameterized gradient descent in the context of sequence models. We showed that the over-parameterization method can adapt to the underlying structure of the signal and significantly outperform the vanilla fixed-eigenvalues method. These results provide a new perspective on the benefits of over-parameterization and offer insights into the adaptivity and generalization properties of neural networks beyond the kernel regime.

However, there are also limitations of this work and many interesting directions for future research. For example, one can directly consider the over-parameterization in the kernel regression by replacing the feature map $\Phi(x) = (\lambda_j^{1/2} e_j(x))_{j \geq 1}$ with the learnable one $\Phi(x; \boldsymbol{a}) = (a_j e_j(x))_{j \geq 1}$, where $a_j$'s are learnable parameters initialized by $a_j(0) = \lambda_j^{1/2}$. However, the analysis would be more challenging since now the components are mutually coupled in the gradient flow dynamics.

Perhaps one of the most interesting directions is to study how the over-parameterization method can also learn the eigenfunctions of the kernel during the training process, which leads to the truly "adaptive kernel method". We believe that future studies on this topic will provide a deeper theoretical understanding of the success of neural networks in practice.

## Acknowledgments and Disclosure of Funding

Qian Lin's research was supported in part by the National Natural Science Foundation of China (Grant 92370122, Grant 11971257).

We thank the anonymous reviewers and area chairs for their valuable comments and suggestions. Their feedback helped us improve the quality of the paper.

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

# Contents

# A Additional related works

In this section, we will provide additional related works and further discussions.

**Regression with fixed kernel** The regression problem with a fixed kernel has been well studied in the literature. It has been shown that with proper regularization, kernel methods can achieve the minimax optimal rates under various conditions [Caponnetto and De Vito, 2007, Steinwart et al., 2009, Lin et al., 2018, Fischer and Steinwart, 2020, Zhang et al., 2023]. Recently, a sequence of works provided more refined results on the generalization error of kernel methods [Li et al., 2023b, Bordelon et al., 2020, Cui et al., 2021, Mallinar et al., 2022, Li et al., 2023a, 2024a].

**The NTK regime of neural networks** Over-parameterized neural networks are connected to kernel methods through the neural tangent kernel (NTK) theory proposed by Jacot et al. [2018], which shows that the dynamics of the neural network at infinite width limited can be approximated by a kernel method with respect to the corresponding NTK. The theory was further developed by many follow-up works [Arora et al., 2019c,b, Du et al., 2018, Lee et al., 2019, Allen-Zhu et al., 2019]. Also, the properties on the corresponding NTK have also been studied [Geifman et al., 2020, Bietti and Bach, 2020, Li et al., 2024b].

**Over-parameterization as Implicit Regularization** There has been a surge of interest in understanding the role of over-parameterization in deep learning. One perspective is that over-parameterized models trained by gradient-based methods can expose certain implicit bias towards simple solutions, which include linear models [Hoff, 2017, Vaškevičius et al., 2019, Zhao et al., 2022, Li et al., 2021a], matrix factorization [Gunasekar et al., 2017, Arora et al., 2019a, Li et al., 2021b, Razin et al., 2021, Chou et al., 2023], linear networks [Yun et al., 2021, Nacson et al., 2022] and neural networks [Kubo et al., 2019, Woodworth et al., 2020]. Moreover, variants of gradient descent such as stochastic gradient descent are also shown to have implicit regularization effects [Li et al., 2022, Vivien et al., 2022]. However, most of these works focus only on the optimization process and the final solution, but the generalization performance is not well understood.

**Generalization Guarantees for Over-parameterized Models** Being the most related to our work, a few works provided generalization guarantees for over-parameterized models, which only include linear models [Zhao et al., 2022, Li et al., 2021a, Vaškevičius et al., 2019] and single index model [Fan et al., 2021]. In detail, the two parallel works [Zhao et al., 2022, Vaškevičius et al., 2019] studied the high-dimensional linear regression problem under sparse settings and showed that a two-layer diagonal over-parameterized model with proper initialization and early stopping can achieve minimax optimal recovery under certain conditions. The subsequent work [Li et al., 2021a] obtained similar results for multi-layer diagonal over-parameterized models.

**The adaptive kernel perspective** The idea of an adaptive kernel has appeared in a few recent works in various forms [Chen et al., 2023, Gatmiry et al., 2021, LeJeune and Alemohammad, 2023, Yang and Hu, 2022, Ba et al., 2022], which is also known as "feature learning". Notably, Gatmiry et al. [2021] showed the benefits brought by the adaptivity of the kernel on a three-layer neural network, which is similar to our work in the adaptive kernel perspective. However, our work and theirs consider different aspects of the adaptive kernel: while they considered an adaptive kernel space in the form of $G \odot K^\infty$ around the NTK space, we consider an eigenvalue-parameterized kernel space with fixed eigen-basis. We believe that these various results, including ours, will contribute to a better understanding of the generalization properties of over-parameterized models as well as neural networks.

# B  Supplementary Technical Details

## B.1  The connection between RKHS and the sequence model

**Diagonalized kernel gradient flow as sequence model**  Moreover, this connection can also be seen directly from the following. The Mercer's decomposition of the RKHS $\mathcal{H}$ associated with the kernel $k$ also provides a series representation of the RKHS:

$$\mathcal{H} = \left\{ \sum_{j=1}^{\infty} a_j \lambda_j^{\frac{1}{2}} e_i \mid (a_j)_{j\geq 1} \in \ell^2 \right\}, \tag{19}$$

where we denote by $\ell^2 = \left\{ (a_j)_{j\geq 1} \mid \sum_{j=1}^{\infty} a_j^2 < \infty \right\}$. Therefore, by introducing the feature mapping $\Phi : \mathcal{X} \to \ell^2$ defined by

$$\Phi(x) = (\lambda_j^{\frac{1}{2}} e_j(x))_{j\geq 1}, \tag{20}$$

we establish a one-to-one correspondence between a function $f \in \mathcal{H}$ in the RKHS and a vector $\beta \in \ell^2$ in feature space via $f(x) = \langle \Phi(x), \beta \rangle_{\ell^2}$ for $f \in \mathcal{H}$ and $\beta \in \ell^2$. Moreover, it is convenient to consider $\Phi(x)$ as column vectors and also define the feature matrix $\Phi(X) = (\Phi(x_1), \ldots, \Phi(x_n))$ for $X = (x_1, \ldots, x_n)$. Then, the gradient flow (2) in the feature space $\ell^2$ writes

$$\dot{\beta}(t) = -\nabla_\beta \mathcal{L} = -\frac{1}{n} \Phi(X)\Phi(X)^\top \beta(t) + \frac{1}{n} \Phi(X)\boldsymbol{y}. \tag{21}$$

Intuitively, since the $j, l$-th entry

$$\left( \frac{1}{n} \Phi(X)\Phi(X)^\top \right)_{j,l} = \frac{\lambda_j^{\frac{1}{2}} \lambda_l^{\frac{1}{2}}}{n} \sum_{i=1}^{n} e_j(x_i) e_l(x_i)$$

the law of large numbers implies that $\frac{1}{n} \Phi(X)\Phi(X)^\top$ converges to the diagonal operator $\Lambda = \mathrm{diag}(\lambda_1, \lambda_2, \ldots)$; moreover, since $j$-th entry

$$\left( \frac{1}{n} \Phi(X)\boldsymbol{y} \right)_j = \frac{\lambda_j^{\frac{1}{2}}}{n} \sum_{i=1}^{n} f(x_i) e_j(x_i) + \frac{\lambda_j^{\frac{1}{2}}}{n} \sum_{i=1}^{n} e_j(x_i)\varepsilon_i,$$

the central limit theorem suggest that it can be approximated by $\lambda_j^{\frac{1}{2}} z_j$, where $z_j = \theta_j + \xi_j$ and $\xi_j$ is a normal random variable with mean zero and variance $\sigma^2/n$. Therefore, with these approximations, the equation can be diagonalized as

$$\dot{\beta}_j(t) = -\lambda_j \beta_j + \lambda_j^{\frac{1}{2}} z_j = -\lambda_j (\beta_j(t) - \lambda_j^{-\frac{1}{2}} z_j).$$

Moreover, we can rewrite the representation $f(x) = \langle \Phi(x), \beta \rangle_{\ell^2}$ into

$$f = \sum_{j=1}^{\infty} \lambda_j^{\frac{1}{2}} \beta_j e_j = \sum_{j=1}^{\infty} \theta_j e_j, \quad \theta_j = \lambda_j^{\frac{1}{2}} \beta_j.$$

Then, in terms of $\theta$, we have

$$\dot{\beta}_j(t) = \lambda_j^{\frac{1}{2}} \dot{\beta}_j(t) = -\lambda_j (\theta_j - z_j).$$

This is exactly the vanilla gradient flow in the sequence model in Section 3.

Furthermore, we can consider the parameterized feature map

$$\Phi_{\boldsymbol{a}}(x) = (a_j e_j(x))_{j\geq 1},$$

where $\boldsymbol{a} = (a_j)_{j\geq 1}$. We can consider similar gradient flow in the feature space with both $\beta$ and $\boldsymbol{a}$ trainable. Then, with similar diagonalizing argument, we can show that the corresponding gradient flow in the sequence model is just the over-parameterized gradient flow in (7). Similar correspondence can be established for the multi-layer models (13).

## B.2 The examples in Section 2

### B.2.1 Example 2.1

The deduction in Example 2.1 is straightforward. The series expansion (3) can be written as

$$[\mathcal{H}_k]^s = \left\{ f = \sum_{j \geq 1} f_j e_j \mid \sum_{j \geq 1} f_j^2 \lambda_j^{-s} < \infty \right\}.$$

We recall that $\lambda_{j,1} \asymp j^{-\gamma_1}$ and $\lambda_{j,2} \asymp j^{-\gamma_2}$. Let $f^* = \sum_{j \geq 1} f_j^* e_j$. Then, for $\ell = 1, 2$,

$$f^* \in [\mathcal{H}_{k_\ell}]^{s_\ell} \iff \sum_{j \geq 1} (f_j^*)^2 \lambda_{j,1}^{-s_\ell} < \infty \iff \sum_{j \geq 1} (f_j^*)^2 j^{\gamma_\ell s_\ell} < \infty$$

Consequently,

$$f^* \in [\mathcal{H}_{k_1}]^{s_1} \iff f^* \in [\mathcal{H}_{k_2}]^{s_2} \quad \text{for} \quad \gamma_1 s_1 = \gamma_2 s_2.$$

### B.2.2 Example 2.2

We justify the claims in Example 2.2. Let us consider the torus $\mathbb{T}^d = [-1, 1)^d$ and the uniform measure $\mu$ on $\mathbb{T}^d$ (namely $\mu(\mathbb{T}^d) = 1$). Let us recall that the multidimensional Fourier basis is given by $\phi_{\boldsymbol{m}}(x) = \exp(i\pi \langle \boldsymbol{m}, x \rangle)$ for $\boldsymbol{m} \in \mathbb{Z}^d$.

The Sobolev space $H^s(\mathbb{T}^d)$ is defined via the Fourier coefficients as

$$H^s(\mathbb{T}^d) = \left\{ f \in L^2(\mathbb{T}^d) \mid \sum_{\boldsymbol{m} \in \mathbb{Z}^d} |f_{\boldsymbol{m}}|^2 (1 + \|\boldsymbol{m}\|^2)^s < \infty \right\},$$

which is equipped with the inner product (as thus the induced norm)

$$\langle f, g \rangle_{H^s(\mathbb{T}^d)} = \sum_{\boldsymbol{m} \in \mathbb{Z}^d} f_{\boldsymbol{m}} \overline{g_{\boldsymbol{m}}} (1 + \|\boldsymbol{m}\|^2)^s.$$

Now, we briefly show that $H^s(\mathbb{T}^d)$ is an RKHS when $s > d/2$. It suffices to show that $f \mapsto f(x)$ is bounded for each $x \in \mathbb{T}^d$ [Andreas Christmann, 2008]. Using the boundedness of $\phi_{\boldsymbol{m}}$, we have

$$\sum_{\boldsymbol{m} \in \mathbb{Z}^d} |f_{\boldsymbol{m}} \phi_{\boldsymbol{m}}| \leq \sum_{\boldsymbol{m} \in \mathbb{Z}^d} |f_{\boldsymbol{m}}| \leq \left[ \sum_{\boldsymbol{m} \in \mathbb{Z}^d} |f_{\boldsymbol{m}}|^2 (1 + \|\boldsymbol{m}\|^2)^s \right]^{\frac{1}{2}} \left[ \sum_{\boldsymbol{m} \in \mathbb{Z}^d} (1 + \|\boldsymbol{m}\|^2)^{-s} \right]^{\frac{1}{2}}$$

Now,

$$\sum_{\boldsymbol{m} \in \mathbb{Z}^d} (1 + \|\boldsymbol{m}\|^2)^{-s} \lesssim \int_{x \in \mathbb{R}^d} (1 + |x|^2)^{-s} dx \lesssim \int_0^\infty (1 + r^2)^{-s} r^{d-1} dr.$$

Since the last integral is finite when $s > d/2$, we find that there is a constant $C$ such that

$$\sum_{\boldsymbol{m} \in \mathbb{Z}^d} |f_{\boldsymbol{m}} \phi_{\boldsymbol{m}}| \leq C \|f\|_{H^s(\mathbb{T}^d)}.$$

Therefore, the series expansion $f(x) = \sum_{\boldsymbol{m} \in \mathbb{Z}^d} f_{\boldsymbol{m}} \phi_{\boldsymbol{m}}(x)$ converges absolutely and uniformly, and thus $|f(x)| \leq \sum_{\boldsymbol{m} \in \mathbb{Z}^d} |f_{\boldsymbol{m}} \phi_{\boldsymbol{m}}| \leq C \|f\|_{H^s(\mathbb{T}^d)}$, showing that $f \mapsto f(x)$ is bounded for each $x \in \mathbb{T}^d$.

Moreover, it is easy to see from the Mercer's decomposition (1) and the power series expansion (3) that the kernel of $H^s(\mathbb{T}^d)$ is given by $k(x, x') = \sum_{\boldsymbol{m} \in \mathbb{Z}^d} (1 + \|\boldsymbol{m}\|^2)^{-s} \phi_{\boldsymbol{m}}(x) \overline{\phi_{\boldsymbol{m}}(x')}$, so its eigenvalues are $\lambda_{\boldsymbol{m}} = (1 + \|\boldsymbol{m}\|^2)^{-s}$.

To determine the eigenvalue decay rate of $\lambda_{\boldsymbol{m}} = (1 + \|\boldsymbol{m}\|^2)^{-s}$ after reordering them in decreasing order, it suffices to determine the count $\#\{\boldsymbol{m} : \lambda_{\boldsymbol{m}} > \delta\}$: the eigenvalue decay rate is $\beta$ if $\#\{\boldsymbol{m} : \lambda_{\boldsymbol{m}} > \delta\} \asymp \delta^{-1/\beta}$, see, e.g., Proposition A.1 in Li et al. [2024a]. We have

$$
\begin{aligned}
\#\{\boldsymbol{m} : \lambda_{\boldsymbol{m}} > \delta\} &= \#\left\{\boldsymbol{m} : (1 + \|\boldsymbol{m}\|^2)^{-s} > \delta\right\} \\
&\asymp \mathrm{Vol}\left(\left\{x \in \mathbb{R}^d : (1 + |x|^2)^{-s} > \delta\right\}\right) \\
&\asymp \mathrm{Vol}\left(\left\{x \in \mathbb{R}^d : |x| < \delta^{-\frac{1}{2s}}\right\}\right) \asymp \delta^{-\frac{d}{2s}}.
\end{aligned}
$$

We consider a function $f^*(x) = g(x_1, \ldots, x_{d_0})$ with low-dimensional structure. Let us denote by $x_{\leq d_0} = (x_1, \ldots, x_{d_0})$ and $x_{> d_0} = (x_{d_0+1}, \ldots, x_d)$ for simplicity. Then, the Fourier coefficients of $f^*$ are given by

$$
\begin{aligned}
f_{\boldsymbol{m}} &= \langle f^*, \phi_{\boldsymbol{m}} \rangle_{L^2(\mathbb{T}, \mathrm{d}\mu)} \\
&= 2^{-d} \int_{\mathbb{T}^{d_0} \times \mathbb{T}^{d-d_0}} g(x_1, \ldots, x_{d_0}) \exp(i\pi \langle \boldsymbol{m}_{\leq d_0}, x_{\leq d_0} \rangle) \cdot \exp(i\pi \langle \boldsymbol{m}_{> d_0}, x_{> d_0} \rangle) \mathrm{d}x_{\leq d_0} \mathrm{d}x_{> d_0} \\
&= 2^{-d_0} \int_{\mathbb{T}^{d_0}} g(x_1, \ldots, x_{d_0}) \exp(i\pi \langle \boldsymbol{m}_{\leq d_0}, x_{\leq d_0} \rangle) \mathrm{d}x_{\leq d_0} \\
&\quad \cdot 2^{-(d-d_0)} \int_{\mathbb{T}^{d-d_0}} \exp(i\pi \langle \boldsymbol{m}_{> d_0}, x_{> d_0} \rangle) \mathrm{d}x_{> d_0} \\
&= g_{\boldsymbol{m}_{\leq d_0}} \cdot \mathbf{1}_{\{\boldsymbol{m}_{> d_0} = \boldsymbol{0}\}},
\end{aligned}
$$

so we show that

$$
f_{\boldsymbol{m}} = \begin{cases} g_{\boldsymbol{m}_{\leq d_0}}, & \boldsymbol{m} = (\boldsymbol{m}_{\leq d_0}, \boldsymbol{0}), \ \boldsymbol{m}_{\leq d_0} \in \mathbb{Z}^{d_0}, \\ 0, & \text{otherwise.} \end{cases}
$$

We recall that $g \in H^t(\mathbb{T}^{d_0})$, so

$$
\sum_{\boldsymbol{m}_{\leq d_0} \in \mathbb{Z}^{d_0}} \left| g_{\boldsymbol{m}_{\leq d_0}} \right|^2 (1 + \|\boldsymbol{m}_{\leq d_0}\|^2)^t < \infty.
$$

To determine the smoothness of $f^*$ on $[\mathcal{H}_{k_1}]^s$ and $[\mathcal{H}_{k_2}]^s$, following (3), we compute

$$
\begin{aligned}
\sum_{\boldsymbol{m} \in \mathbb{Z}^d} |f_{\boldsymbol{m}}|^2 \left[(1 + \|\boldsymbol{m}\|^2)^r\right]^s &= \sum_{\boldsymbol{m} = (\boldsymbol{m}_{\leq d_0}, \boldsymbol{0}), \boldsymbol{m}_{\leq d_0} \in \mathbb{Z}^{d_0}} \left| g_{\boldsymbol{m}_{\leq d_0}} \right|^2 \left[(1 + \|\boldsymbol{m}_{\leq d_0}\|^2)^r\right]^s \\
&\qquad \sum_{\boldsymbol{m}_{\leq d_0} \in \mathbb{Z}^{d_0}} \left| g_{\boldsymbol{m}_{\leq d_0}} \right|^2 (1 + \|\boldsymbol{m}_{\leq d_0}\|^2)^{rs},
\end{aligned}
$$

so $f^*$ belongs to $[\mathcal{H}_{k_1}]^s$ and $[\mathcal{H}_{k_2}]^s$ for $s = t/r$

### B.2.3 Example 2.3

We recall that $f^* = \sum_{j \geq 1} \theta_j^* e_j$,

$$
\left| \theta_{l(j)}^* \right| \asymp j^{-(p+1)/2} \quad \text{and} \quad \ell(j) \asymp j^q \quad \text{for} \quad p > 0, \ q \geq 1,
$$

where $\ell(j)$ gives the descending order of $\left| \theta_j^* \right|$. To compute the relative smoothness of $f^*$ w.r.t. $\lambda_{j,1} \asymp j^{-\gamma}$, we compute

$$
\sum_{j \geq 1} \left| \theta_j^* \right|^2 \lambda_j^{-s} = \sum_{j \geq 1} \left| \theta_{l(j)}^* \right|^2 \lambda_{l(j)}^{-s} \asymp \sum_{j \geq 1} j^{-(p+1)} (\ell(j))^{\gamma s} \asymp \sum_{j \geq 1} j^{-(p+1)} j^{q\gamma s} \asymp \sum_{j \geq 1} j^{-1-(p-q\gamma s)},
$$

so we have $s < p/(q\gamma)$ (but arbitrarily close) and the corresponding generalization error rate is $\frac{s\gamma}{s\gamma+1} < \frac{p}{p+q}$. The generalization error rate w.r.t. $\lambda_{l(j),2} \asymp j^{-\gamma}$ can be computed similarly.

## C   Detailed Numerical Experiments

In this section, we provide numerical experiments to validate the theoretical results. The codes are provided in the supplementary material. We approximate the gradient flow equation (22) and (30) by discrete-time gradient descent with sufficiently small step size. Moreover, we truncate the sequence model to the first $N$ terms for some very large $N$. We consider the settings as in Corollary 3.3 that $\boldsymbol{\theta}^*$ is given by (4) for some $p > 0$ and $q \geq 1$. We set $\epsilon^2 = n^{-1}$, where $n$ can be regarded as the sample size, and consider the asymptotic performance of the generalization error as $n$ grows. For the stopping time, we choose the oracle one that minimizes the generalization error for each method. We first compare the generalization error rates between vanilla gradient descent and over-parameterized gradient descent (OpGD) in Figure 3 on page 21. The results show that the over-parameterized gradient descent can achieve the optimal generalization error rate, while the vanilla gradient descent suffers from the misalignment caused by $q > 1$ and thus has a slower convergence rate. Moreover, with a logarithmic least-squares fitting, we find that the resulting generalization error rates are consistent with the theoretical results in Corollary 3.3 (0.5 for OpGD and 0.33 for vanilla GD); the oracle stopping times for the over-parameterized gradient descent also match the theoretical value (0.5).

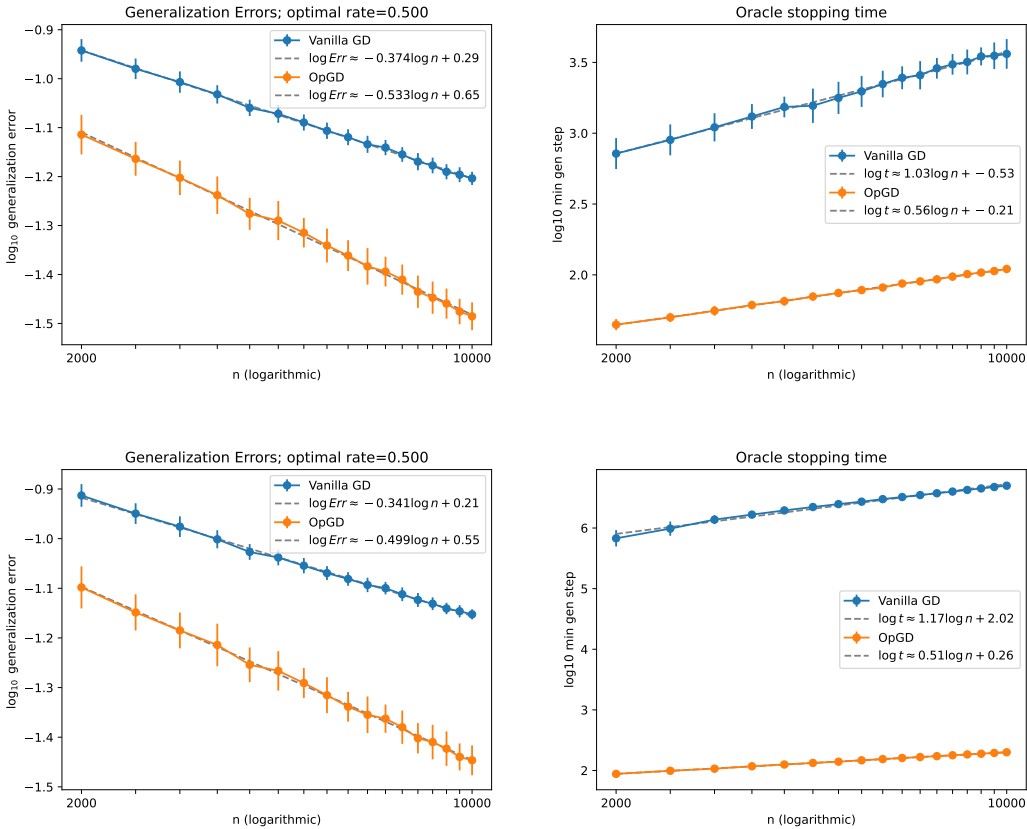

Figure 3: Comparison of the generalization error rates between vanilla gradient descent and over-parameterized gradient descent (OpGD). We set $p = 1$ and $q = 2$ for the truth parameter $\boldsymbol{\theta}^*$. The left and right columns show respectively the generalization error and the orcale stopping time with respect to $n$. For the upper row, we set the eigenvalue decay rate $\gamma = 1.5$; for the lower row, we set $\gamma = 3$. For each $n$, we repeat 64 times and plot the mean and the standard deviation.

Furthermore, we investigate the generalization performance of over-parameterized gradient descent (also with deeper parameterization) for different settings of the truth parameter $\boldsymbol{\theta}^*$, the eigenvalue decay rate $\gamma$ and the depth $D$. The results are reported in Table C on page 22. We find that the generalization error rates are in general consistent with the theoretical results in Corollary 3.3, while

|  | $p=0.6\ (r^*=0.37)$ | | | $p=1\ (r^*=0.5)$ | | | $p=3\ (r^*=0.75)$ | | |
|---|---|---|---|---|---|---|---|---|---|
| $\gamma$ | $q=1$ | $q=1.5$ | $q=2$ | $q=1$ | $q=1.5$ | $q=2$ | $q=1$ | $q=1.5$ | $q=2$ |
| 1.1 | 0.38 | 0.40 | 0.50 | 0.50 | 0.45 | 0.48 | 0.78 | 0.69 | 0.67 |
| 2 | 0.36 | 0.41 | 0.52 | 0.49 | 0.46 | 0.50 | 0.80 | 0.73 | 0.72 |
| 3 | 0.36 | 0.41 | 0.52 | 0.48 | 0.46 | 0.50 | 0.76 | 0.73 | 0.74 |

Table 1: Convergence rates of the over-parameterized gradient descent (8) under different settings of the truth parameter $p, q$ and the eigenvalue decay rate $\gamma$, where $r^*$ is the ideal convergence rate. The convergence rate is estimated by the logarithmic least-squares fitting of the generalization error with $n$ ranging from $2000, 2200, \dots, 4000$, where the generalization error is the mean of 256 repetitions.

|  |  | $p=0.6\ (r^*=0.37)$ | | | $p=1\ (r^*=0.5)$ | | | $p=3\ (r^*=0.75)$ | | |
|---|---|---|---|---|---|---|---|---|---|---|
|  | $\gamma$ | $q=1$ | $q=1.5$ | $q=2$ | $q=1$ | $q=1.5$ | $q=2$ | $q=1$ | $q=1.5$ | $q=2$ |
|  | 1.1 | 0.36 | 0.40 | 0.52 | 0.49 | 0.46 | 0.49 | 0.79 | 0.73 | 0.72 |
| $D=1$ | 2 | 0.36 | 0.40 | 0.52 | 0.48 | 0.46 | 0.50 | 0.76 | 0.73 | 0.73 |
|  | 3 | 0.30 | 0.32 | 0.38 | 0.45 | 0.41 | 0.44 | 0.75 | 0.73 | 0.74 |
|  | 1.1 | 0.34 | 0.39 | 0.49 | 0.46 | 0.44 | 0.48 | 0.76 | 0.74 | 0.75 |
| $D=2$ | 2 | 0.35 | 0.40 | 0.51 | 0.47 | 0.45 | 0.49 | 0.74 | 0.73 | 0.73 |
|  | 3 | 0.36 | 0.40 | 0.51 | 0.48 | 0.46 | 0.50 | 0.74 | 0.73 | 0.73 |

Table 2: Convergence rates of the over-parameterized gradient descent (14) with $D=1$ and $D=2$. The settings are the same as in Table C on page 22.

there are some fluctuations due to the finite sample size. Comparing the results for $\gamma=1.1$ across depth $D=0$, $D=1$ and $D=2$, we see that the method with $D=0$ has the slowest convergence rate, while the method with $D=2$ has the fastest convergence rate, justifying that deeper parameterization can improve the generalization performance. In summary, the numerical experiments validate our theoretical results.

Additionally, we investigate the evolution of the eigenvalue terms $a_j(t)b_j^D(t)$ over time $t$ as discussed in Proposition 3.4. The results are shown in Figure 4 on page 23. We find that the eigenvalue terms can indeed adapt to the underlying structure of the signal: for large signals, the eigenvalue terms approach the signals as the training progresses, while for small signals, the eigenvalue terms do not increase significantly. Moreover, we find that deeper over-parameterization reduces the fluctuations of the eigenvalue terms for the noise components, and thus improves the generalization performance of the model.

### C.1 Experiments beyond the sequence model

We also explore the adaptivity of the over-parameterized gradient descent beyond the sequence model. Let us consider the diagonal adaptive kernel method by parameterizing the feature map with $\Phi(x; \boldsymbol{a}) = (a_j e_j(x))_{j \geq 1}$ introduced in Section 5. We use the setting in Example 2.2 where the eigenfunctions are the trigonometric functions. In particular, we consider the truth function $f^*(x) = \sin(7.5\pi x_1)$ with $x = (x_1, x_2) \in \mathbb{R}^2$. We present the generalization error curve of a single trial and also the generalization error rates with respect to the sample size $n$ in Figure 5 on page 24. The result also shows the benefit of over-parameterization in adapting to the underlying structure of the signal.

### C.2 Testing eigenvalue misalignment on real-world data

In this section, we provide additional experiments to test the eigenvalue misalignment phenomenon on real-world data. Recalling Example 2.2 and Example 2.3, we know that the misalignment happens when the order of the eigenvalues of the kernel mismatches the order of coefficients of the truth function. Therefore, to test the misalignment, we compute the coefficients of the regression function over the eigen-basis of the kernel and examine whether the coefficients decay in the order given by the kernel. For the eigen-basis, we use the multidimensional Fourier basis (the trigonometric functions) considered in Example 2.2, where the order is given by the descending order of $\lambda_{\boldsymbol{m}} = (1 + \|\boldsymbol{m}\|^2)^{-r}$.

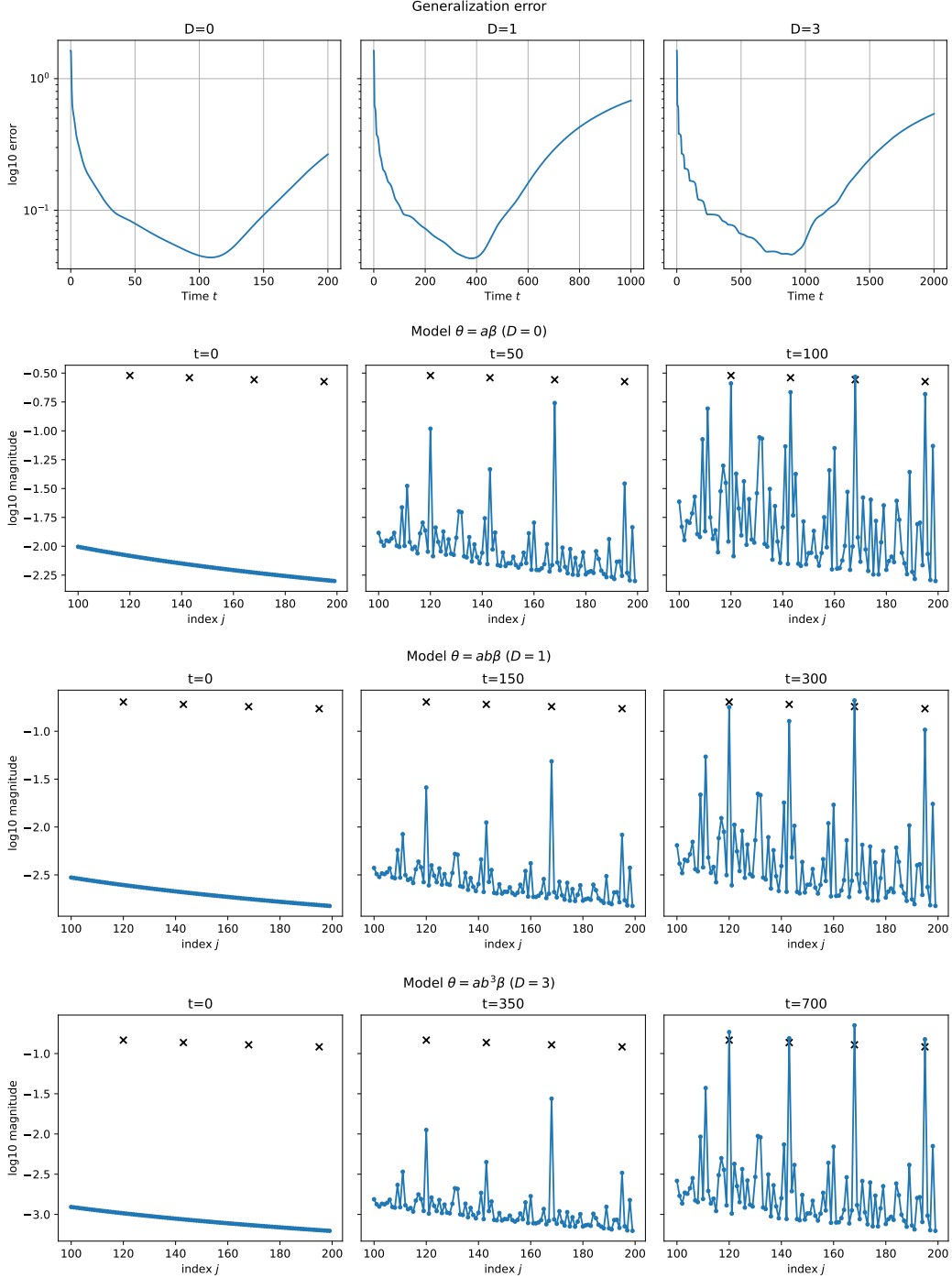

Figure 4: The generalization error as well as the evolution of the eigenvalue terms $a_j(t)b_j^D(t)$ over the time $t$. The first row shows the generalization error of three parameterizations $D = 0, 1, 3$ with respect to the training time $t$. The rest of the rows show the evolution of the eigenvalue terms $a_j(t)b_j^D(t)$ over the time $t$. For presentation, we select the index $j = 100$ to $200$. The blue line shows the eigenvalue terms and the black marks show the non-zero signals scaled according to Proposition 3.4. For the settings, we set $p = 1$, $q = 2$ and $\gamma = 2$.

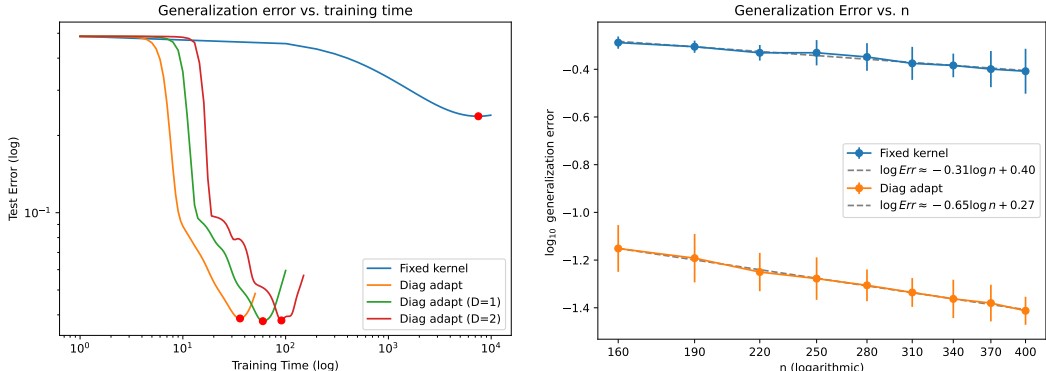

Figure 5: Comparison of the generalization error between the fixed kernel gradient method and the diagonal adaptive kernel method. The left figure shows the generalization error curve of a single trial. The right figure shows the generalization error rates with respect to the sample size $n$.

We consider the two real-world datasets: "California Housing" and "Concrete Compressive Strength". We compute the empirical inner product of the regression function with the Fourier basis functions up to a certain order. Then, we plot the coefficients in the order given by the kernel. The results are shown in Figure 6 on page 25. The figures show that the empirical coefficients exhibit significant spikes. Also, among the coefficients, only very few components have large magnitudes, indicating the sparse structure of the regression function. Together, these results suggest that the eigenvalues of the kernel are misaligned with the truth function in these datasets.

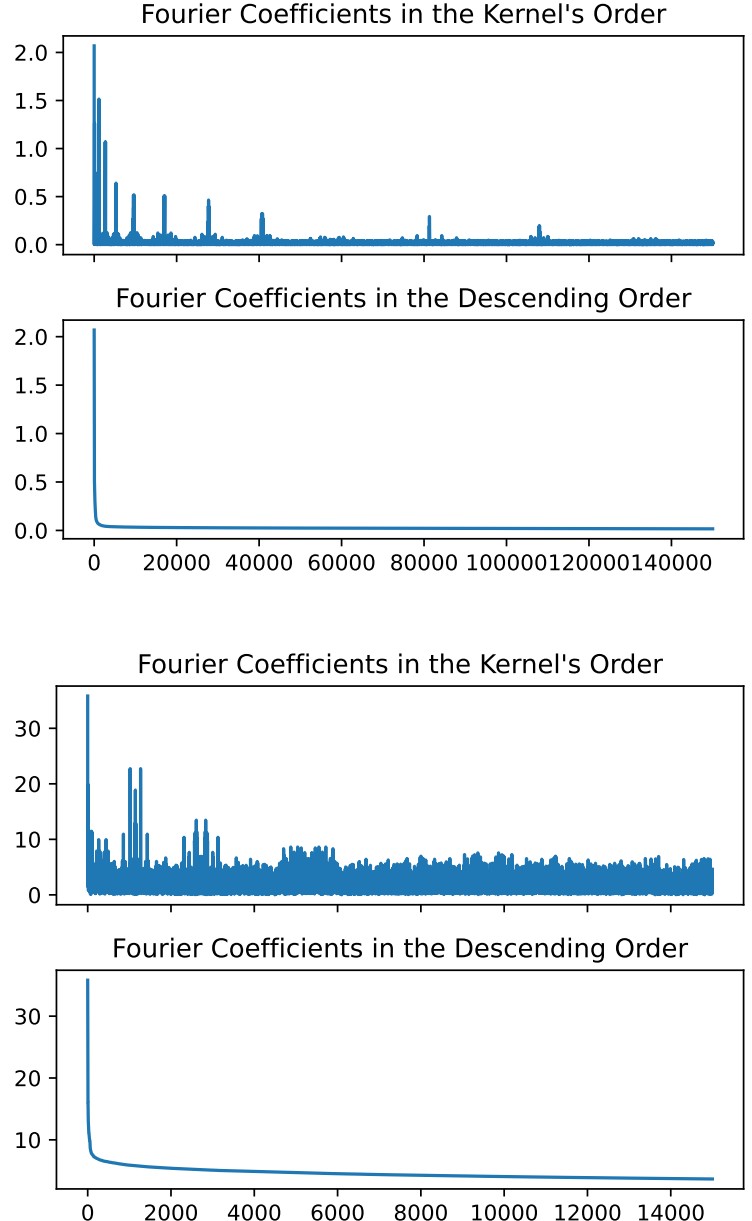

Figure 6: The empirical coefficients of the regression function over the Fourier basis for the "California Housing" dataset (upper) and "Concrete Compressive Strength" dataset (lower). Note that we take different numbers of Fourier basis functions for the two datasets for better visualization.

# D   Analysis of the gradient flow

## D.1   The two-layer parameterization

Let us consider the gradient flow considered in (8), where we remove the subscript $j$ for notational simplicity. Let $L = \frac{1}{2}(\theta - z)^2$ and $\theta = a\beta$. We are interested in the gradient flow:

$$\begin{aligned}
\dot{a} &= -\nabla_a L = \beta(z - \theta), \\
\dot{\beta} &= -\nabla_\beta L = a(z - \theta), \\
a(0) &= \lambda^{\frac{1}{2}} > 0, \quad \beta(0) = 0.
\end{aligned} \tag{22}$$

**Symmetry of the solution**   We can find that the solution of the equation for $z < 0$ can be obtained by simply $a(t), -\beta(t)$ for the positive signal case of $-z > 0$. Therefore, we only need to consider the case of $z > 0$. In this case, it is obvious that $a(t)$, $\beta(t)$ and $\theta(t)$ are all non-negative and increasing.

**Gradient flow of $\theta$**   Now we notice that

$$\frac{1}{2}\frac{\mathrm{d}}{\mathrm{d}t}a^2 = \frac{1}{2}\frac{\mathrm{d}}{\mathrm{d}t}\beta^2 = a\beta(z - \theta),$$

so

$$a^2(t) - \beta^2(t) \equiv a^2(0) - \beta^2(0) = \lambda. \tag{23}$$

Using this conservation quantity, we can prove the following estimations:

$$\begin{aligned}
\theta &= a\beta = \sqrt{\lambda + \beta^2} \cdot \beta \geq \beta^2, \\
\theta &= a\beta = a\sqrt{a^2 - \lambda} \leq a^2.
\end{aligned} \tag{24}$$

Moreover, the derivative of $\theta$ writes

$$\dot{\theta} = a\dot{\beta} + \dot{a}\beta = (a^2 + \beta^2)(z - \theta).$$

Using $a^2 + \beta^2 = \sqrt{(a^2 - \beta^2)^2 + 4a^2\beta^2} = \sqrt{\lambda^2 + 4\theta^2}$, we conclude the follow explicit equation for $\theta$:

$$\dot{\theta} = \sqrt{\lambda^2 + 4\theta^2}(z - \theta). \tag{25}$$

Then, we have the following approximation of the solution.

**Lemma D.1.** *Let us consider the gradient flow* (25) *and*

$$\frac{\mathrm{d}}{\mathrm{d}t}\tilde{\theta} = (\lambda + 2|\tilde{\theta}|)(z - \tilde{\theta}), \quad \tilde{\theta}(0) = 0. \tag{26}$$

*Then we have*

$$\begin{aligned}
0 \leq \tilde{\theta}(t/\sqrt{2}) \leq \theta(t) \leq \tilde{\theta}(t) \leq z \quad &\text{if} \quad z \geq 0; \\
0 \geq \tilde{\theta}(t/\sqrt{2}) \geq \theta(t) \geq \tilde{\theta}(t) \geq z \quad &\text{if} \quad z \leq 0.
\end{aligned} \tag{27}$$

*Moreover,* (26) *is solved by*

$$\tilde{\theta}(t) = \frac{\lambda(E - 1)}{2|z| + \lambda E}z, \qquad E = \exp((2|z| + \lambda)t). \tag{28}$$

*Proof.* It suffices to consider the case $z \geq 0$. It is easy to see from the gradient flow (22) that both $a(t), \beta(t)$ are non-negative. Then, using the elementary inequality

$$\frac{1}{\sqrt{2}}(\lambda + 2x) \leq \sqrt{\lambda^2 + 4x^2} \leq \lambda + 2x,$$

we have

$$\frac{1}{\sqrt{2}}(\lambda + 2x)(z - x) \leq \sqrt{\lambda^2 + 4x^2}(z - x) \leq (\lambda + 2x)(z - x).$$

Then, the comparison principal in ordinary differential equation yields (27). The verification of (28) is straightforward. $\qquad\square$

## D.2 Deeper parameterization

Now let us consider the deeper parameterization of the form

$$\theta = ab^D\beta, \tag{29}$$

where $a, b, \beta$ are all trainable parameters, and the gradient flow

$$\begin{aligned}
\dot{a} &= -\nabla_a L = b^D\beta(z-\theta), \\
\dot{b} &= -\nabla_b L = Dab^{D-1}\beta(z-\theta), \\
\dot{\beta} &= -\nabla_\beta L = ab^D(z-\theta), \\
a(0) &= \lambda^{\frac{1}{2}} > 0, \quad b(0) = b_0 > 0, \quad \beta(0) = 0.
\end{aligned} \tag{30}$$

**Main idea** We provide the main idea of analyzing the gradient flow (30) here. Using the conservation quantities, we can first focus on the equation of $\beta$. For the initial stage when $t$ is relatively small, $\beta(t)$ only grows linearly in $t$. Next, when $\beta(t)$ exceeds a certain threshold depending on the initialization (and also the interplay between $\lambda_j$ and $b_0$), $\beta(t)$ grows exponentially in $t$, provided that $\theta(t) \leq z/2$. Thus, we can upper bound the hitting time of $\theta(t)$ to $z/2$. Finally, when $\theta(t) \geq z/2$, we consider directly the equation of $\theta$ and show that $\theta(t)$ converges to $z$ exponentially fast.

**Symmetry of the solution** Similar to the two-layer case, we can find that the solution of the equation for $z < 0$ can be obtained by negating the sign of $\beta(t)$ for the positive signal case. Therefore, we will focus on the case of $z \geq 0$ where $a(t), b(t), \beta(t)$ and thus $\theta(t)$ are all non-negative and non-decreasing.

**Conservation quantities** Similarly, we have

$$\frac{1}{2}\frac{\mathrm{d}}{\mathrm{d}t}a^2 = \frac{1}{2D}\frac{\mathrm{d}}{\mathrm{d}t}b^2 = \frac{1}{2}\frac{\mathrm{d}}{\mathrm{d}t}\beta^2 = \theta(z-\theta). \tag{31}$$

so

$$a = \left(\lambda + \beta^2\right)^{\frac{1}{2}}, \quad b = \left(b_0^2 + D\beta^2\right)^{\frac{1}{2}}. \tag{32}$$

Using these conservation quantities, we can prove the following estimations in terms of $\beta$:

$$\begin{aligned}
\min(\lambda^{\frac{1}{2}}, \beta) &\leq a \leq \sqrt{2}\max(\lambda^{\frac{1}{2}}, \beta), \\
\min(b_0, \sqrt{D}\beta) &\leq b \leq \sqrt{2}\max(b_0, \sqrt{D}\beta).
\end{aligned} \tag{33}$$

**The evolution of $\theta$** It is direct to compute that

$$\begin{aligned}
\dot{\theta} &= \dot{a}b^D\beta + aDb^{D-1}\dot{b}\beta + ab^D\dot{\beta} \\
&= \left[(b^D\beta)^2 + (Dab^{D-1}\beta)^2 + (ab^D)^2\right](z-\theta) \\
&= \theta^2(a^{-2} + Db^{-2} + \beta^{-2})(z-\theta).
\end{aligned} \tag{34}$$

**Auxiliary notations** Let us introduce

$$T^{(1)} = \inf\left\{t \geq 0 : \beta(t) \geq \lambda^{\frac{1}{2}}\right\}, \quad T^{(2)} = \inf\left\{t \geq 0 : \beta(t) \geq b_0/\sqrt{D}\right\}, \tag{35}$$

$$T^{\mathrm{esc}} = \min(T^{(1)}, T^{(2)}), \quad T^{\mathrm{sig}} = \inf\left\{t \geq 0 : \theta(t) \geq z/2\right\}.$$

**Lemma D.2** (Noise case). *For the gradient flow* (30)*, we have the initial estimation*

$$\begin{aligned}
|\beta(t)| &\leq 2^{\frac{D+1}{2}}\lambda^{\frac{1}{2}}b_0^D|z|t, \\
|\theta(t)| &\leq 2^{D+1}\lambda b_0^{2D}|z|t,
\end{aligned} \quad \text{for} \quad t \leq \min(\underline{T}^{(1)}, \underline{T}^{(2)}), \tag{36}$$

*where*

$$\underline{T}^{(1)} = \left(2^{\frac{D+1}{2}}b_0^D|z|\right)^{-1}, \quad \underline{T}^{(2)} = \left(2^{\frac{D+1}{2}}\sqrt{D}\lambda^{\frac{1}{2}}b_0^{D-1}|z|\right)^{-1}. \tag{37}$$

*Moreover, if $\lambda^{\frac{1}{2}} \leq b_0/\sqrt{D}$, then*

$$|\beta(t)| \leq \lambda^{\frac{1}{2}} \exp\left(2^{\frac{D+1}{2}} b_0^D |z|(t - \underline{T}^{(1)})^+\right),$$
$$|\theta(t)| \leq 2^{\frac{D+1}{2}} \lambda b_0^D \exp\left(2^{\frac{D+3}{2}} b_0^D |z|(t - \underline{T}^{(1)})^+\right), \qquad for \quad t \leq \underline{T}^{(1,2)}, \qquad (38)$$

*where*

$$\underline{T}^{(1,2)} = \left(1 + \ln \frac{b_0}{\sqrt{D}\lambda^{\frac{1}{2}}}\right) \underline{T}^{(1)}. \qquad (39)$$

*Proof.* It suffices to consider the case $z > 0$. Recalling (30) and $\theta \geq 0$, we have

$$\dot\beta \leq ab^D z.$$

Using the upper bound in (33), when $t \leq T^{\text{esc}}$, namely when $\beta(t) \leq \lambda^{\frac{1}{2}}$ and $\sqrt{D}\beta(t) \leq b_0$, we have

$$\dot\beta \leq (\sqrt{2}\lambda^{\frac{1}{2}})(\sqrt{2}b_0)^D z = 2^{\frac{D+1}{2}} \lambda^{\frac{1}{2}} b_0^D z,$$

implying that

$$\beta(t) \leq 2^{\frac{D+1}{2}} \lambda^{\frac{1}{2}} b_0^D zt, \quad for \quad t \leq T^{\text{esc}}.$$

Therefore, we get the lower bound

$$T^{\text{esc}} \geq \min(\underline{T}^{(1)}, \underline{T}^{(2)}),$$

where $\underline{T}^{(1)}$ and $\underline{T}^{(2)}$ are defined by (37) in the lemma. Combining this again with the upper bound that $\theta = ab^D \beta \leq 2^{\frac{D+1}{2}} a_0 b_0^D \beta$ when $t \leq T^{\text{esc}}$, we prove (36).

For the second part, we consider the case $\lambda^{\frac{1}{2}} \leq b_0/\sqrt{D}$. In this case, we have $T^{(1)} \leq T^{(2)}$ and thus $T^{(1)} \geq \underline{T}^{(1)}$ from the above argument. Now, when $t \in [T^{(1)}, T^{(2)}]$, we turn to the following equation:

$$\dot\beta \leq (\sqrt{2}\beta)(\sqrt{2}b_0)^D z = 2^{\frac{D+1}{2}} b_0^D z\beta, \quad for \quad t \in [T^{(1)}, T^{(2)}],$$

which yields

$$\beta(s + T^{(1)}) \leq \beta(T^{(1)}) \exp\left(2^{\frac{D+1}{2}} b_0^D zs\right) = \lambda^{\frac{1}{2}} \exp\left(2^{\frac{D+1}{2}} b_0^D zs\right), \quad for \quad s \in [0, T^{(2)} - T^{(1)}].$$

Comparing $\beta(s + T^{(1)})$ with $b_0/\sqrt{D}$ gives

$$T^{(2)} - T^{(1)} \geq \left[2^{\frac{D+1}{2}} b_0^D z\right]^{-1} \ln \frac{b_0}{\sqrt{D}\lambda^{\frac{1}{2}}} = \underline{T}^{(1)} \ln \frac{b_0}{\sqrt{D}\lambda^{\frac{1}{2}}},$$

so

$$T^{(2)} \geq T^{(1)} + \underline{T}^{(1)} \ln \frac{b_0}{\sqrt{D}\lambda^{\frac{1}{2}}} \geq \underline{T}^{(1)} \left(1 + \ln \frac{b_0}{\sqrt{D}\lambda^{\frac{1}{2}}}\right) = \underline{T}^{(1,2)}.$$

Therefore, the comparison principal yields

$$\beta(t) \leq \lambda^{\frac{1}{2}} \exp\left(2^{\frac{D+1}{2}} b_0^D |z|(t - \underline{T}^{(1)})^+\right) \quad for \quad t \leq \underline{T}^{(1,2)},$$

where we notice that the bound also holds for $t \leq \underline{T}^{(1)} \leq T^{(1)}$ since at that time $\beta(t) \leq \lambda^{\frac{1}{2}}$. Finally, (38) is obtained by using $\theta = ab^D \beta \leq 2^{\frac{D+1}{2}} b_0^D \beta^2$ when $t \in [T^{(1)}, T^{(2)}]$, while the bound also holds for $t \leq T^{(1)}$.

$\qquad\qquad\qquad\qquad\qquad\qquad\qquad\qquad\qquad\qquad\qquad\qquad\qquad\qquad\qquad\qquad\qquad\qquad\qquad\square$

**Lemma D.3** (Signal case). *For the gradient flow* (30), *we have:*

- *If $\lambda^{\frac{1}{2}} \leq b_0/\sqrt{D}$, then*

$$T^{\text{sig}} \leq 2(b_0^D |z|)^{-1} \left[1 + \left(\ln \frac{(D^{-D/2} z/2)^{\frac{1}{D+2}}}{\lambda^{\frac{1}{2}}}\right)^+\right], \qquad (40)$$

- *If $\lambda^{\frac{1}{2}} \geq b_0/\sqrt{D}$, then*

$$T^{\mathrm{sig}} \leq 2 \left( \sqrt{D} \lambda^{\frac{1}{2}} b_0^{D-1} |z| \right)^{-1} \left( 1 + R^+ \right), \tag{41}$$

    *where*

$$R = \begin{cases} \ln \frac{(D|z|/2)^{\frac{1}{D+2}}}{b_0}, & D = 1, \\ \frac{1}{D-1}, & D > 1. \end{cases}$$

*Moreover, we have*

$$|z - \theta(t)| \leq \frac{1}{2} |z| \exp\left( -\frac{1}{4} D^{\frac{D}{D+2}} |z|^{\frac{2D+2}{D+2}} (t - T^{\mathrm{sig}}) \right), \quad for \quad t \geq T^{\mathrm{sig}}. \tag{42}$$

*Proof.* It suffices to consider the case $z > 0$. To provide an upper bound of the signal time $T^{\mathrm{sig}}$, we observe that the lower bound in (33) implies a sufficient condition for $\theta \geq z/2$ that

$$\beta \geq \left( D^{-D/2} z/2 \right)^{\frac{1}{D+2}} \quad \Longrightarrow \quad \theta \geq \frac{1}{2} z. \tag{43}$$

We first consider case that $\lambda^{\frac{1}{2}} \leq b_0/\sqrt{D}$. Let us define $\overline{T}^{(1)} := 2(b_0^D z)^{-1}$ and suppose that $T^{\mathrm{sig}} \geq 2(b_0^D z)^{-1}$, otherwise the statement (40) already holds. Then, we first have

$$\dot{\beta} = ab^D(z - \theta) \geq \frac{1}{2} \lambda^{\frac{1}{2}} b_0^D z, \quad for \quad t \leq T^{\mathrm{sig}}. \tag{44}$$

This implies that

$$\beta(t) \geq \frac{1}{2} \lambda^{\frac{1}{2}} b_0^D z t, \quad for \quad t \leq T^{\mathrm{sig}},$$

and thus

$$T^{(1)} \leq \overline{T}^{(1)} \leq T^{\mathrm{sig}}.$$

Now, for $t \in [T^{(1)}, T^{\mathrm{sig}}]$, we use the alternative bound $a \geq \beta$ to obtain

$$\dot{\beta} \geq \frac{1}{2} \beta b_0^D z, \quad for \quad t \in [T^{(1)}, T^{\mathrm{sig}}],$$

giving that

$$\beta(s + T^{(1)}) \geq \lambda^{\frac{1}{2}} \exp\left( \frac{1}{2} b_0^D z s \right), \quad for \quad s \in [0, T^{\mathrm{sig}} - T^{(1)}].$$

Comparing it with (43), we obtain

$$T^{\mathrm{sig}} - T^{(1)} \leq 2(b_0^D z)^{-1} \ln \frac{(D^{-D/2} z/2)^{\frac{1}{D+2}}}{\lambda^{\frac{1}{2}}} = \overline{T}^{(1)} \ln \frac{(D^{-D/2} z/2)^{\frac{1}{D+2}}}{\lambda^{\frac{1}{2}}},$$

which, together with the upper bound of $T^{(1)}$, proves (40).

The case that $\lambda^{\frac{1}{2}} \geq b_0/\sqrt{D}$ is very similar. We define $\overline{T}^{(2)} := 2(\sqrt{D} \lambda^{\frac{1}{2}} b_0^{D-1} z)^{-1}$ and suppose that $T^{\mathrm{sig}} \geq \overline{T}^{(2)}$. Using the estimation (44) again, we get

$$T^{(2)} \leq \overline{T}^{(2)} \leq T^{\mathrm{sig}}.$$

Now, when $t \in [T^{(2)}, T^{\mathrm{sig}}]$, we use $b \geq \sqrt{D}\beta$ to obtain

$$\dot{\beta} \geq \frac{1}{2} \lambda^{\frac{1}{2}} D^{\frac{D}{2}} \beta^D z, \quad for \quad t \in [T^{(2)}, T^{\mathrm{sig}}].$$

This implies that for $s \leq T^{\mathrm{sig}} - T^{(2)}$,

$$\beta(s + T^{(2)}) \geq \frac{b_0}{\sqrt{D}} \exp\left( \frac{1}{2} \lambda^{\frac{1}{2}} D^{\frac{D}{2}} z s \right), \quad if \quad D = 1,$$

$$\beta(s + T^{(2)}) \geq \left[ (D^{-1/2} b_0)^{-(D-1)} - \frac{D-1}{2} \lambda^{\frac{1}{2}} D^{\frac{D}{2}} zs \right]^{-\frac{1}{D-1}}, \quad \text{if} \quad D > 1.$$

Consequently, comparing it with (43) gives

$$T^{\text{sig}} - T^{(2)} \leq 2 \left( \lambda^{\frac{1}{2}} D^{\frac{D}{2}} z \right)^{-1} \ln \frac{(Dz/2)^{\frac{1}{D+2}}}{b_0} = \overline{T}^{(2)} \ln \frac{(Dz/2)^{\frac{1}{D+2}}}{b_0}, \quad \text{if} \quad D = 1,$$

$$T^{\text{sig}} - T^{(2)} \leq \frac{2}{D-1} \left( \sqrt{D} \lambda^{\frac{1}{2}} b_0^{D-1} z \right)^{-1} = \frac{1}{D-1} \overline{T}^{(2)}, \quad \text{if} \quad D > 1.$$

Finally, let us consider the convergence stage when $t \geq T^{\text{sig}}$. Since it is always true that $\theta \leq z$, using the lower bounds in (32), we have

$$z \geq \theta = ab^D \beta \geq \beta \cdot D^{\frac{D}{2}} \beta^D \cdot \beta = D^{\frac{D}{2}} \beta^{D+2},$$

implying that $\beta \leq D^{-\frac{D}{2(D+2)}} z^{\frac{1}{D+2}}$. Now, plugging this into (34) and noticing that $\theta \geq z/2$ when $t \geq T^{\text{sig}}$, we derive

$$\begin{aligned}
\dot{\theta} &= \theta^2 (a^{-2} + Db^{-2} + \beta^{-2})(z - \theta) \\
&\geq \theta^2 \beta^{-2}(z - \theta) \\
&\geq \frac{1}{4} z^2 D^{\frac{D}{D+2}} z^{-\frac{2}{D+2}} (z - \theta) \\
&= \frac{1}{4} D^{\frac{D}{D+2}} z^{\frac{2D+2}{D+2}} (z - \theta).
\end{aligned}$$

Therefore, we have

$$z - \theta(s + T^{\text{sig}}) \leq (z - \theta(T^{\text{sig}})) \exp \left( -\frac{1}{4} D^{\frac{D}{D+2}} z^{\frac{2D+2}{D+2}} s \right) = \frac{1}{2} \exp \left( -\frac{1}{4} D^{\frac{D}{D+2}} z^{\frac{2D+2}{D+2}} s \right),$$

$\square$

# E  Main proofs

**Notations**  For notation simplicity, we will use $C, c$ to denote generic positive constants that may change from line to line.

## E.1  Proof of Theorem 3.1

We recall that

$$\mathbb{E}\mathcal{R}(\hat{\boldsymbol{\theta}}; \boldsymbol{\theta}^*) = \mathbb{E}\sum_{j=1}^{\infty}(\hat{\theta}_j - \theta_j^*)^2.$$

Let us define the signal event

$$S_j = \left\{\omega : |\xi_j| < \frac{1}{2}|\theta_j^*|\right\}, \quad S_j^{\complement} = \left\{\omega : |\xi_j| \geq \frac{1}{2}|\theta_j^*|\right\}. \tag{45}$$

Then, on $S_j$ we have $\frac{1}{2}|\theta_j^*| \leq |z_j| \leq \frac{3}{2}|\theta_j^*|$, while on $S_j^{\complement}$ we have $|z_j| \leq 3|\xi_j|$. Then, we decompose

$$(\hat{\theta}_j - \theta_j^*)^2 = (\hat{\theta}_j - \theta_j^*)^2\mathbf{1}_{S_j} + (\hat{\theta}_j - \theta_j^*)^2\mathbf{1}_{S_j^{\complement}}.$$

Moreover, when the signal is significant, we use

$$(\hat{\theta}_j - \theta_j^*)^2\mathbf{1}_{S_j} \leq 2(\hat{\theta}_j - z_j)^2\mathbf{1}_{S_j} + 2(z_j - \theta_j^*)^2\mathbf{1}_{S_j} = 2(\hat{\theta}_j - z_j)^2\mathbf{1}_{S_j} + 2\xi_j^2\mathbf{1}_{S_j}.$$

On the other hand, when the noise is dominating, we apply

$$(\hat{\theta}_j - \theta_j^*)^2\mathbf{1}_{S_j^{\complement}} \leq 2\hat{\theta}_j^2\mathbf{1}_{S_j^{\complement}} + 2(\theta_j^*)^2\mathbf{1}_{S_j^{\complement}}.$$

Summing over $j$ and taking the expectation, we have

$$\mathcal{R}(\hat{\boldsymbol{\theta}}^{\mathrm{Op}}; \boldsymbol{\theta}^*) = \mathbb{E}\sum_{j=1}^{\infty}(\hat{\theta}_j - \theta_j^*)^2 = \mathbb{E}\sum_{j=1}^{\infty}(\hat{\theta}_j - \theta_j^*)^2\mathbf{1}_{S_j} + \mathbb{E}\sum_{j=1}^{\infty}(\hat{\theta}_j - \theta_j^*)^2\mathbf{1}_{S_j^{\complement}}$$

$$\leq 2\mathbb{E}\sum_{j=1}^{\infty}(\hat{\theta}_j - z_j)^2\mathbf{1}_{S_j} + 2\mathbb{E}\sum_{j=1}^{\infty}\xi_j^2\mathbf{1}_{S_j^{\complement}} + 2\mathbb{E}\sum_{j=1}^{\infty}\hat{\theta}_j^2\mathbf{1}_{S_j^{\complement}} + 2\mathbb{E}\sum_{j=1}^{\infty}(\theta_j^*)^2\mathbf{1}_{S_j^{\complement}}$$

$$= 2\mathbb{E}\sum_{j=1}^{\infty}\left[\xi_j^2\mathbf{1}_{S_j} + (\theta_j^*)^2\mathbf{1}_{S_j^{\complement}}\right] \tag{46}$$

$$+ 2\mathbb{E}\sum_{j=1}^{\infty}\hat{\theta}_j^2\mathbf{1}_{S_j^{\complement}} \tag{47}$$

$$+ 2\mathbb{E}\sum_{j=1}^{\infty}(\hat{\theta}_j - z_j)^2\mathbf{1}_{S_j}. \tag{48}$$

Now, the first term (46), representing the absolute error, is controlled by Proposition E.1 that

$$\mathbb{E}\sum_{j=1}^{\infty}\left[\xi_j^2\mathbf{1}_{S_j} + (\theta_j^*)^2\mathbf{1}_{S_j^{\complement}}\right] \leq 4\left[\epsilon^2\Phi(\epsilon) + \Psi(\epsilon)\right].$$

Therefore, we focus on the remaining two terms and obtain the estimations (49) and (51) in the following.

**The noise term**  The term (47) represents the extra error caused by the estimator when the noise dominates. Applying Lemma D.1, we obtain

$$\left|\hat{\theta}_j\right| = |\theta_j(t)| \leq \left|\tilde{\theta}(t)\right| = \frac{\lambda_j(E_j - 1)}{2|z_j| + \lambda_j E_j}|z_j| \leq \frac{1}{2}\lambda_j\exp\left((6|\xi_j| + \lambda_j)t\right) \quad \text{on} \quad S_j^{\complement}.$$

Let us choose $J = \min\{j \geq 1 : \lambda_j \leq \epsilon\} \asymp \epsilon^{-1/\gamma}$. Then, since $t \leq B_2\epsilon^{-1}$, we have $(|\xi_j| + \lambda_j)t \leq B_2(|\xi_j|/\epsilon + 1)$ and thus

$$\mathbb{E}\sum_{j \geq J}\hat{\theta}_j^2\mathbf{1}_{S_j^{\complement}} \leq C\sum_{j \geq J}\lambda_j^2\mathbb{E}\exp(C(|\xi_j|/\epsilon) + C) \leq C\sum_{j \geq J}\lambda_j^2 \leq CJ^{-(2\gamma-1)} \leq C\epsilon^{2-1/\gamma},$$

where we notice that $\mathbb{E}\exp(C(|\xi_j|/\epsilon) + C)$ is uniformly bounded by some constant for all $j$ since each $|\xi_j|/\epsilon$ is 1-sub-Gaussian. On the other hand, using the obvious bound $\left|\hat{\theta}_j\right| \leq 3|\xi_j|$ on $S_j^{\complement}$, we obtain

$$\mathbb{E}\sum_{j < J}\hat{\theta}_j^2\mathbf{1}_{S_j^{\complement}} \leq C\sum_{j < J}\mathbb{E}\xi_j^2 \leq C\epsilon^2 J \leq C\epsilon^{2-1/\gamma}.$$

Combining two terms, we conclude that

$$\mathbb{E}\sum_{j=1}^{\infty}\hat{\theta}_j^2\mathbf{1}_{S_j^{\complement}} \leq C\epsilon^{2-1/\gamma}. \tag{49}$$

**The signal term**    The term (47) represents the approximation error when the signal is significant. We apply Lemma D.1 again to derive

$$\left|\hat{\theta}_j - z_j\right| \leq \left|\tilde{\theta}(t/\sqrt{2}) - z_j\right| = \frac{2|z_j| + \lambda_j}{2|z_j| + \lambda_j\exp\big((2|z_j| + \lambda_j)t/\sqrt{2}\big)}|z_j|.$$

Using the fact that $\frac{1}{2}\left|\theta_j^*\right| \leq |z_j| \leq \frac{3}{2}\left|\theta_j^*\right|$ on $S_j$, we derive that

$$(\hat{\theta}_j - z_j)^2\mathbf{1}_{S_j} \leq C\frac{(\left|\theta_j^*\right| + \lambda_j)^2\theta_j^2}{\lambda_j^2\exp\big((2\left|\theta_j^*\right| + \lambda_j)t/\sqrt{2}\big)}. \tag{50}$$

Let us define $\nu = \epsilon\ln(1/\epsilon) \geq \epsilon$. Recalling (9) and using Assumption 1, we have

$$j \leq \max J_{\text{sig}}(\epsilon) \leq C\epsilon^{-\kappa}, \quad \text{for} \quad j \in J_{\text{sig}}(\nu) \subseteq J_{\text{sig}}(\epsilon).$$

Now, if we take $t \geq B_1\epsilon^{-1}$ for some constant $B_1$, since $\left|\theta_j^*\right| \geq \nu$ for $j \in J_{\text{sig}}(\nu)$, we also have

$$\frac{1}{\sqrt{2}}\left|\theta_j^*\right|t \geq \frac{1}{\sqrt{2}}t\epsilon\ln(1/\epsilon) \geq cB_1\ln(1/\epsilon),$$

and thus when $j \in J_{\text{sig}}(\nu)$,

$$\ln\left[\lambda_j^2\exp\left(\left|\theta_j^*\right|t/\sqrt{2}\right)\right] = 2\ln\lambda_j + \frac{1}{\sqrt{2}}\left|\theta_j^*\right|t \geq cB_1\ln(1/\epsilon) - C\ln j \geq (cB_1 - C)\ln(1/\epsilon).$$

Consequently, as long as $B_1$ is large enough, we have

$$\lambda_j^2\exp\left(\left|\theta_j^*\right|t/\sqrt{2}\right) \geq 1 \quad \text{when} \quad j \in J_{\text{sig}}(\nu).$$

Therefore, plugging this into (50), we get

$$(\hat{\theta}_j - z_j)^2\mathbf{1}_{S_j} \leq C\frac{(\left|\theta_j^*\right| + \lambda_j)^2\theta_j^2}{\lambda_j^2\exp\big((2\left|\theta_j^*\right| + \lambda_j)t/\sqrt{2}\big)} \leq C\exp\left(-(\left|\theta_j^*\right| + \lambda_j)t/(\sqrt{2})\right)(\left|\theta_j^*\right| + \lambda_j)^2\theta_j^2,$$

and hence

$$\sum_{j \in J_{\text{sig}}(\nu)}\mathbb{E}(\hat{\theta}_j - z_j)^2\mathbf{1}_{S_j} \leq \sum_{j \in J_{\text{sig}}(\nu)}\exp\left(-(\left|\theta_j^*\right| + \lambda_j)t/(\sqrt{2})\right)(\left|\theta_j^*\right| + \lambda_j)^2\theta_j^2$$

$$\leq C\sum_{j=1}^{\infty}\left[(\left|\theta_j^*\right| + \lambda_j)t\right]^{-2}(\left|\theta_j^*\right| + \lambda_j)^2\theta_j^2$$

$$= Ct^{-2}\sum_{j=1}^{\infty}\theta_j^2 \leq C\epsilon^2.$$

On the other hand, with the trivial bound $\left|\hat{\theta}_j - z_j\right| \leq |z_j|$, the remaining terms are bounded by

$$\sum_{j \notin J_{\text{sig}}(\nu)} \mathbb{E}(\hat{\theta}_j - z_j)^2 \mathbf{1}_{S_j} \leq \sum_{j \notin J_{\text{sig}}(\nu)} (\theta_j^*)^2 = \Psi(\nu).$$

Therefore, we conclude that

$$\mathbb{E} \sum_{j=1}^{\infty} (\hat{\theta}_j - z_j)^2 \mathbf{1}_{S_j} \leq C\epsilon^2 + \Psi\left(\epsilon \ln(1/\epsilon)\right). \tag{51}$$

## E.2    Proof of Theorem 3.2

Following the same argument as the proof in the previous section, we introduce the events $S_j$ and $S_j^{\complement}$ in (45) and decompose the error as in (46), (47) and (48). Then, we will focus on the last two terms and derive the estimations (53) and (54) in the following. We recall that $b_0 \asymp \epsilon^{\frac{1}{D+2}}$ and we choose $t$ such that $B_1 \epsilon^{-1} \leq b_0^D t \leq B_2 \epsilon^{-1}$ for some constants $B_1, B_2 > 0$ that will be determined later.

Here, we also note that the in correspondence to the component-wise gradient flow considered in Subsection D.2, the initializations are given by $\lambda = \lambda_j$ and $b_{0,j} = b_0$ in (30). Now, since $\lambda_j \asymp j^{-\gamma}$, the index

$$J = \min \left\{ j \geq 1 : \lambda_j^{1/2} \leq b_{0,j}/\sqrt{D} \right\} \asymp b_0^{-2/\gamma}. \tag{52}$$

**The noise term**    To apply the bounds in Lemma D.2, let us denote the event

$$A_j = \left\{ \omega : 3 \cdot 2^{\frac{D+1}{2}} b_0^D |\xi_j| t \leq \ln \frac{b_0}{\lambda_j^{\frac{1}{2}} \sqrt{D}} \right\}.$$

Then, since $|z_j| \leq 3|\xi_j|$ on $S_j^{\complement}$, we have $t \leq \underline{T}_j^{(1,2)}$ on $A_j$, where $\underline{T}_j^{(1,2)}$ is defined via (39). Then, for $j > J$, applying (38) yields

$$\hat{\theta}_j^2 \mathbf{1}_{S_j^{\complement} \cap A_j} \leq 2^{D+1} b_0^{2D} \lambda_j^2 \exp\left(2^{\frac{D+5}{2}} b_0^D |z_j| t\right) \mathbf{1}_{S_j^{\complement} \cap A_j}$$

$$\leq 2^{D+1} b_0^{2D} \lambda_j^2 \exp\left(2^{\frac{D+5}{2}} B_2 |z_j|/\epsilon\right) \mathbf{1}_{S_j^{\complement} \cap A_j}$$

$$\leq C b_0^{2D} \lambda_j^2 \exp(C|\xi_j|/\epsilon) \mathbf{1}_{S_j^{\complement} \cap A_j}$$

where we use $b_0^D t \leq B_2 \epsilon^{-1}$ in the last inequality. Consequently,

$$\mathbb{E} \sum_{j>J} \hat{\theta}_j^2 \mathbf{1}_{S_j^{\complement} \cap A_j} \leq C b_0^{2D} \sum_{j>J} \lambda_j^2 \mathbb{E} \exp(C|\xi_j|/\epsilon) \leq C b_0^{2D} \sum_{j>J} \lambda_j^2$$

$$\leq C b_0^{2D} J^{-(2\gamma-1)} \leq C b_0^{2(D+2-1/\gamma)},$$

where in the second inequality we notice that $\mathbb{E} \exp(C|\xi_j|/\epsilon)$ is uniformly bounded since each $|\xi_j|/\epsilon$ is 1-sub-Gaussian.

On the other hand, noticing $b_0^D t \leq B_2 \epsilon^{-1}$ again, we have

$$j \in A_j^{\complement} \implies C b_0^D t |\xi_j| \geq \ln \frac{b_0}{\lambda_j^{\frac{1}{2}} \sqrt{D}}$$

$$\implies |\xi_j|/\epsilon \geq c B_2^{-1} \ln \frac{b_0}{\lambda_j^{\frac{1}{2}} \sqrt{D}} = c B_2^{-1} \ln\left(b_0^2 \lambda_j^{-1}/D\right).$$

Hence, using Lemma F.1 with the sub-Gaussian property of $\xi_j$ and noticing that $b_0^2 \lambda_j^{-1}/D \geq 1$ when $j > J$, we obtain

$$\mathbb{E} \sum_{j>J} \hat{\theta}_j^2 \mathbf{1}_{S_j^{\complement} \cap A_j^{\complement}} \leq C \sum_{j>J} \mathbb{E}\left[\xi_j^2 \mathbf{1}\left\{|\xi_j|/\epsilon \geq c B_2^{-1} \ln\left(b_0^2 \lambda_j^{-1}/D\right)\right\}\right]$$

$$\leq C\epsilon^2 \sum_{j>J} \exp\left(-c\left[cB_2^{-1}\ln(b_0^2\lambda_j^{-1}/D)\right]^2\right)$$

$$\leq C\epsilon^2 \sum_{j>J} \exp\left(-c\left[\ln(b_0^2 j^\gamma)\right]^2\right)$$

$$\leq C\epsilon^2 \int_J^\infty \exp\left(-c\left[\ln(b_0^2 x^\gamma)\right]^2\right)\mathrm{d}x$$

$$\leq C\epsilon^2 b_0^{-2/\gamma} \int_c^\infty \exp\left(-c\left[\ln(y^\gamma)\right]^2\right)\mathrm{d}y, \quad y = b_0^{2/\gamma}x$$

$$\leq C\epsilon^2 b_0^{-2/\gamma} \leq C\epsilon^2 \epsilon^{-\frac{2}{D+2}\frac{1}{\gamma}}.$$

Finally, using the bound $\left|\hat{\theta}_j\right| \leq 3|\xi_j|$ again, the remaining terms are bounded by

$$\mathbb{E}\sum_{j\leq J} \hat{\theta}_j^2 \mathbf{1}_{S_j^{\complement}} \leq \epsilon^2 J \leq C\epsilon^2 b_0^{-2/\gamma} \leq C\epsilon^2 \epsilon^{-\frac{2}{D+2}\frac{1}{\gamma}}.$$

In summary, we conclude that

$$\mathbb{E}\sum_{j=1}^\infty \hat{\theta}_j^2 \mathbf{1}_{S_j^{\complement}} \leq C\epsilon^2 \epsilon^{-\frac{2}{D+2}\frac{1}{\gamma}}. \tag{53}$$

**The signal term** We will apply the bound in Lemma D.3. Let us denote

$$J_{\text{rec}} = \left\{j : t \geq 2T_j^{\text{sig}}\right\} \cap J_{\text{sig}}(\epsilon).$$

Then, when $j \in J_{\text{rec}}$ and $S_j$ holds, (42) and the fact $\frac{1}{2}|\theta_j^*| \leq |z_j| \leq \frac{3}{2}|\theta_j^*|$ imply

$$|\theta_j - z_j| \leq \frac{1}{2}|z_j|\exp\left(-\frac{1}{4}D^{\frac{D}{D+2}}z^{\frac{2D+2}{D+2}}(t - T_j^{\text{sig}})\right) \leq C|\theta_j^*|\exp\left(-c|\theta_j^*|^{\frac{2D+2}{D+2}}t\right).$$

Consequently,

$$\mathbb{E}\sum_{j\in J_{\text{rec}}} (\hat{\theta}_j - z_j)^2 \mathbf{1}_{S_j} \leq C\sum_{j\in J_{\text{rec}}} (\theta_j^*)^2 \exp\left(-c|\theta_j^*|^{\frac{2D+2}{D+2}}t\right) \leq C\sum_{j\in J_{\text{rec}}} (\theta_j^*)^2 \left(|\theta_j^*|^{\frac{2D+2}{D+2}}t\right)^{-\frac{D+2}{D+1}}$$

$$= C\sum_{j\in J_{\text{rec}}} t^{-\frac{D+2}{D+1}} \leq Ct^{-\frac{D+2}{D+1}}|J_{\text{sig}}(\epsilon)| \leq C\epsilon^2\Phi(\epsilon),$$

where we use $\exp(-cx) \leq Cx^{-\frac{D+2}{D+1}}$ in the second inequality.

Let us define $\nu = \epsilon\ln(1/\epsilon) \geq \epsilon$. We claim that $J_{\text{rec}}^{\complement} \subseteq J_{\text{sig}}(\nu)^{\complement}$ on $S_j$ as long as $b_0^D t \geq B_1\epsilon^{-1}$ for some large constant $B_1$. Then, using the obvious bound $\left|\hat{\theta}_j - z_j\right| \leq |z_j| \leq \frac{3}{2}|\theta_j^*|$ on $S_j$, we have

$$\mathbb{E}\sum_{j\in J_{\text{rec}}^{\complement}} (\hat{\theta}_j - z_j)^2 \mathbf{1}_{S_j} \leq \sum_{j\in J_{\text{sig}}(\nu)^{\complement}} (\theta_j^*)^2 = \Psi(\nu). \tag{54}$$

To prove the claim, we show that $J_{\text{sig}}(\nu) \subseteq J_{\text{rec}}$ on $S_j$ as long as $B_1$ is large enough. Recalling (9) and using Assumption 1, for $j \in J_{\text{sig}}(\nu) \subseteq J_{\text{sig}}(\epsilon)$, we have

$$j \leq \max J_{\text{sig}}(\epsilon) \leq C\epsilon^{-\kappa}.$$

Now, we show that $t \geq 2T_j^{\text{sig}}$ for $j \in J_{\text{sig}}(\nu)$ on $S_j$ for different cases in Lemma D.3.

- If $\lambda_j^{1/2} \leq b_0/\sqrt{D}$, we have (40) and thus

$$t \geq 2T_j^{\text{sig}} \quad \Longleftarrow \quad b_0^D|z_j|t \geq 1 + \left(\ln\frac{(D^{-D/2}z/2)^{\frac{1}{D+2}}}{\lambda_j^{1/2}}\right)^+$$

$$\Longleftarrow \quad \frac{B_1}{2}|\theta_j|\epsilon^{-1} \geq C\ln\left(|\theta_j^*|\lambda_j^{-1}\right) + C$$

$$\Longleftarrow \quad B_1\epsilon\ln(1/\epsilon)\epsilon^{-1} \geq C\gamma\ln j + C$$

$$\Longleftarrow \quad B_1\ln(1/\epsilon) \geq C\kappa\ln(1/\epsilon) + C.$$

- If $\lambda_j^{1/2} \geq b_0/\sqrt{D}$, (41) gives

$$t \geq 2T_j^{\text{sig}} \quad \Longleftarrow \quad \sqrt{D}\lambda_j^{1/2}b_0^{D-1}|z_j|t \geq 1 + R_j^+,$$

where

$$R_j = \begin{cases} \ln\dfrac{(D|z_j|/2)^{\frac{1}{D+2}}}{b_0}, & D = 1, \\ \dfrac{1}{D-1}, & D > 1. \end{cases}$$

So for both $D = 1$ and $D > 1$, we have similarly

$$t \geq 2T_j^{\text{sig}} \quad \Longleftarrow \quad \frac{1}{2}\sqrt{D}\lambda_j^{1/2}b_0^{D-1}|\theta_j^*|t \geq C\ln\left(b_0^{-1}\right) + C$$

$$\Longleftarrow \quad \frac{1}{2}|\theta_j|b_0^D t \geq C\ln(1/\epsilon) + C$$

$$\Longleftarrow \quad B_1\ln(1/\epsilon) \geq C\ln(1/\epsilon) + C.$$

Therefore, for both cases, we have $t \geq 2T_j^{\text{sig}}$ as long as $B_1$ is large enough. This finishes the proof of the claim.

### E.3 The absolute error term

The following proposition connect the absolute error term with the ideal risk in Johnstone [2017].

**Proposition E.1.** For the sequence model (5), recalling the signal events (45) and the quantities (9), we have

$$\mathbb{E}\sum_{j=1}^{\infty}\left[\xi_j^2\mathbf{1}_{S_j} + \theta_j^2\mathbf{1}_{S_j^\complement}\right] \leq 4\sum_{j=1}^{\infty}\min(\epsilon^2, \theta_j^2) = 4\left[\epsilon^2\Phi(\epsilon) + \Psi(\epsilon)\right]. \tag{55}$$

*Proof.* It is straightforward to see that

$$\xi_j^2\mathbf{1}_{S_j} + \theta_j^2\mathbf{1}_{S_j^\complement} = \xi_j^2\mathbf{1}_{\{2|\xi_j|<|\theta_j|\}} + \theta_j^2\mathbf{1}_{\{2|\xi_j|\geq|\theta_j|\}} \leq \min(4\xi_j^2, \theta_j^2),$$

so

$$\mathbb{E}\left[\xi_j^2\mathbf{1}_{S_j} + \theta_j^2\mathbf{1}_{S_j^\complement}\right] \leq \mathbb{E}\min(4\xi_j^2, \theta_j^2) \leq 4\mathbb{E}\min(\xi_j^2, \theta_j^2) \leq 4\min(\epsilon^2, \theta_j^2).$$

Summing over $j$ yields the inequality. The last equality follows from the definition of $\Phi(\epsilon)$ and $\Psi(\epsilon)$. $\qquad\square$

### E.4 Proof of Proposition 3.4

The fact that $a_j(t)b_j^D(t)$ is non-decreasing follows from the analysis of the gradient flow in Subsection D.1 and Subsection D.2. For $\delta \in (0,1)$, let us choose $C$ large enough such that $\mathbb{P}\{|\xi_j| \leq C\epsilon\} \geq 1 - \delta$ for any fixed $j$.

**The case $D = 0$**   For the signal component where $|\theta_j^*| \geq 2C\epsilon\ln(1/\epsilon)$, we have $|z_j| \geq \frac{1}{2}|\theta_j^*|$ with high probability. Then, we follow the analysis of the signal term in Subsection E.1 and obtain that

$$|\theta_j(t) - z_j|^2 \leq C(\theta_j^*)^2 t^{-2} \leq \frac{1}{4}(\theta_j^*)^2,$$

provided that $\epsilon$ is small enough. This implies that

$$|\theta_j(t)| \geq \frac{1}{4}|\theta_j^*|.$$

Now, the second inequality in (24) implies that $|\theta_j(t)| \leq a_j^2(t)$. Consequently, we conclude that $a_j(t) \geq \frac{1}{2}|\theta_j^*|^{1/2}$.

For the noise component where $|\theta_j^*| \leq \epsilon$, we have $|z_j| \leq (C+1)\epsilon$ with high probability. Moreover, since $\lambda_j \asymp j^{-\gamma}$, we have $J = \min\{j \geq 1 : \lambda_j \leq \epsilon\} \asymp \epsilon^{-1/\gamma}$. Following the similar analysis of the noise term in Subsection E.1, when $j \geq C\epsilon^{-1/\gamma}$ for some $C > 0$, we have

$$|\theta_j(t)| \leq \lambda_j \exp(C(|z_j| + \lambda_j)t) \leq \lambda_j \exp(C\epsilon t) \leq C\lambda_j.$$

Then, the first inequality in (24) gives $|\theta_j(t)| \geq \beta_j^2(t)$, so we have $|\beta_j(t)| \leq C\lambda_j^{1/2}$. Finally,

$$a_j(t) = \sqrt{\lambda_j + \beta_j^2(t)} \leq \sqrt{\lambda_j + C\lambda_j} \leq C\lambda_j^{1/2}.$$

**The case $D \geq 1$**   For the signal component where $|\theta_j^*| \geq 2C\epsilon\ln(1/\epsilon)$, we still have we have $|z_j| \geq \frac{1}{2}|\theta_j^*|$ with high probability. Now, from the analysis of the signal term in Subsection E.2, we have $t \geq 2T_j^{\text{sig}}$. Moreover, investigating the proof of Lemma D.3, we see that the analysis in Subsection E.2 actually shows that

$$|\beta_j(t)| \geq c|z_j|^{\frac{1}{D+2}} \geq c|\theta_j^*|^{\frac{1}{D+2}}.$$

Consequently, (32) implies that

$$a_j(t)b_j^D(t) \geq |\beta_j(t)|^{D+1} \geq c|\theta_j^*|^{\frac{D+1}{D+2}}.$$

For the noise component where $|\theta_j^*| \leq \epsilon$, we also have $|z_j| \leq (C+1)\epsilon$ with high probability. Now, we have

$$|z_j|b_0^D t \leq (C+1)\epsilon b_0^D t \leq C_0,$$

for some constant $C_0$. Following the similar analysis of the noise term in Subsection E.2, we can choose $j \geq C\epsilon^{-\frac{2}{D+2}\frac{1}{\gamma}}$ such that

$$1 + \ln\frac{b_0}{\lambda_j^{1/2}\sqrt{D}} \geq C_0.$$

Now, this condition guarantees that $t \leq \underline{T}^{(1,2)}$ defined in Lemma D.2, so Lemma D.2 gives

$$|\beta(t)| \leq \lambda_j^{\frac{1}{2}}\exp\left(Cb_0^D|z_j|(t - \underline{T}^{(1)})^+\right) \leq \lambda_j^{\frac{1}{2}}\exp\left(Cb_0^D|z_j|t\right) \leq C\lambda_j^{\frac{1}{2}}.$$

Combining it with the upper bound in (32) and noticing $t \leq \underline{T}^{(1,2)}$ yield

$$a_j(t)b_j^D(t) \leq 2^{\frac{D+1}{2}}|\beta_j(t)|b_0^D \leq C\epsilon^{\frac{D}{D+2}}\lambda_j^{\frac{1}{2}}.$$

## F  Auxiliary results

**Lemma F.1.** *Suppose $X$ is $\sigma^2$-sub-Gaussian, namely, $\mathbb{P}\left\{|X| \geq t\right\} \leq 2\exp\left(-\frac{1}{2\sigma^2}t\right)$ for $t \geq 0$. Then for $M \geq 0$, we have the tail bound*

$$\mathbb{E}X^2 \mathbf{1}\left\{|X| \geq M\right\} \leq 4\sigma^2 \exp\left(-\frac{1}{4\sigma^2}M^2\right). \tag{56}$$

*Proof.* Using integration by parts, we have

$$\begin{aligned}
\mathbb{E}X^2 \mathbf{1}\left\{X \geq M\right\} &= 2\int_0^\infty r\mathbb{P}\left\{|X| \geq \max(M, r)\right\}\mathrm{d}r \\
&\leq 4\int_0^\infty r\exp\left(-\frac{1}{2\sigma^2}\max(M^2, r^2)\right)\mathrm{d}r \\
&= \left(M^2 + 2\sigma^2\right)\exp\left(-\frac{M^2}{2\sigma^2}\right) \\
&\leq 4\sigma^2\exp\left(-\frac{1}{4\sigma^2}M^2\right).
\end{aligned}$$

$\square$

**Lemma F.2.** *Suppose that $(\theta_j)_{j\geq 1}$ satisfies $\left|\theta_{l(j)}\right| \asymp j^{-(p+1)/2}$ for some $p > 0$ and $|\theta_j| = 0$ otherwise, where $l(j)$ is a sequence of indices. Defining $\Phi(\delta)$ and $\Psi(\delta)$ as in (9), we have*

$$\Phi(\delta) \asymp \delta^{-\frac{2}{p+1}}, \qquad \Psi(\delta) \asymp \delta^{\frac{2p}{p+1}}.$$

*Proof.* First, from the definition of $\Phi(\delta)$ and $\Psi(\delta)$, we see that they do not depend on ordering of the indices and zero values of $\theta_j$. Therefore, we can assume that $l(j) = j$. Then, assuming that $c_1 j^{-(p+1)/2} \leq |\theta_j| \leq C_1 j^{-(p+1)/2}$, we have

$$\Phi(\delta) = |\{j : |\theta_j| \geq \delta\}| \leq \left|\left\{j : C_1 j^{-(p+1)/2} \geq \delta\right\}\right| \leq (\delta/C_1)^{-\frac{2}{p+1}}.$$

Moreover,

$$\begin{aligned}
\Psi(\delta) &= \sum_{j=1}^\infty |\theta_j|^2 \mathbf{1}\left\{|\theta_j| < \delta\right\} \\
&= \sum_{j>\Phi(\delta)} |\theta_j|^2 \leq C_1^2 \sum_{j>\Phi(\delta)} j^{-(p+1)} \\
&\leq C_1^2 C\Phi(\delta)^{-p} \leq C'\delta^{\frac{2p}{p+1}}
\end{aligned}$$

for some constant $C' > 0$. The lower bound of them can be obtained similarly. $\square$

