# OpenReview forum: "Improving Adaptivity via Over-Parameterization in Sequence Models"
_NeurIPS.cc/2024/Conference — NeurIPS 2024 poster_

### Official Review · Reviewer_cuCj · 2024-06-22

**Soundness:** 3
**Presentation:** 2
**Contribution:** 3
**Rating:** 5
**Confidence:** 2

**Summary:**

This paper investigates the influence of over-parameterization on the adaptivity and generalization of sequence models. The work highlights the significance of eigenfunctions in kernel regression and introduces an over-parameterized gradient descent method to analyze the effects of varying eigenfunction orders. Theoretical results demonstrate that over-parameterized models can adapt to the underlying signal structure and outperform traditional methods. The research shows that deeper over-parameterization further enhances model generalization. This approach provides insights into improving the flexibility and performance of neural networks, especially in practical scenarios where network architecture and initialization conditions dynamically evolve.

**Strengths:**

The paper presents a novel approach by leveraging over-parameterization in sequence models to enhance adaptivity and generalization. This originality is evident in its creative combination of kernel regression techniques and over-parameterized gradient descent methods, which have not been explored together in this context before. Its significance lies transforming the understanding of neural network adaptivity beyond the traditional NTK framework, providing a robust theoretical foundation for improving model performance in practical scenarios where network architecture and initialization are dynamic.

**Weaknesses:**

One significant weakness of the paper is the limited experimental validation. While the theoretical results are compelling, the empirical evidence provided is not sufficient to fully support the claims made about the superiority of the over-parameterized models in practical scenarios. The paper would benefit from a more extensive set of experiments across various datasets and model architectures to demonstrate the robustness and generalizability of the proposed methods. Additionally, there is a need for more practical examples that illustrate how the approach can be applied in real-world settings, especially given the dynamic nature of network architecture and initialization conditions mentioned. Including these aspects would enhance the paper's contributions and practical relevance.

**Questions:**

1. Can you provide more detailed theoretical justification or insights into why deeper layers contribute to better performance? Specifically, how does the introduction of additional layers influence the adaptation of eigenvalues, and are there any diminishing returns or optimal depth considerations that should be taken into account?

2. The theoretical results are promising, but the empirical validation is limited. Have you considered testing the proposed over-parameterized gradient descent method on a wider range of datasets and model architectures? How does the method perform in different domains, such as natural language processing or computer vision, compared to traditional methods and other over-parameterization techniques?

---

> ### Author Rebuttal · Authors · 2024-08-06
>
> Thank you for your efforts in reviewing our paper and your comments.
> We appreciate that you recognize that "the theoretical results are compelling".
> We would like to address your concerns in the following:
>
> > One significant weakness of the paper is the limited experimental validation.
> >
> > 2.The theoretical results are promising, but the empirical validation is limited.
>
>
> We appreciate the reviewer’s concern about the experimental validation.
>
> First, we would like to point out that while our experiments are mostly done on the sequence model,
> **similar results hold for the kernel model under the non-parametric regression setting.**
> We have performed additional experiments and we will include them in the new revision of the paper.
>
> More importantly, the goal of our paper is to capture the dynamic nature of neural networks under the kernel framework
> and investigate the **impact of the dynamic evolution of the kernel on the generalization performance**.
> The reason why we consider a simplified model is that (1) directly analyzing a deep neural network is technically
> significantly challenging;
> (2) the simplified model enables us to gain a clear understanding of the dynamic evolution of the kernel and its impact
> on generalization.
> Moreover, the experiments included in our paper, particularly those in Appendix A, align well with our theoretical
> results.
> Therefore, our simplified model allows us to **gain insights and provides a solid theoretical foundation for
> understanding more complicated over-parameterized models**.
>
> From a higher perspective, our main insight is that over-parameterized models combined with gradient-based training
> methods lead to a dynamically adaptive kernel that can adapt to the structure of the truth signal,
> which we would like to refer to as the *"adaptive kernel" perspective*.
> This perspective would be **a valuable stepping stone for understanding the generalization properties of more
> complex neural networks**.
> Under the guidance of this perspective, we will be able to consider more complicated over-parameterization and more
> realistic models like fully connected deep neural networks.
>
> Therefore, while our current model may not directly apply to real-world datasets in natural language processing or
> computer vision, we think that our insights gained in this simplified setting will guide further research that explores
> the practical implications of our findings in these and other areas.
> We believe that starting from this point, we will be able to consider more realistic settings and understand more
> complex models in the near future.
>
> We hope this clarification addresses your concern and highlights the intended scope of our work.
>
>
>
>
>
> > 1.Can you provide more detailed theoretical justification or insights into why deeper layers contribute to better
> > performance? Specifically, how does the introduction of additional layers influence the adaptation of eigenvalues, and
> > are there any diminishing returns or optimal depth considerations that should be taken into account?
>
> Thank you for the insightful question. To summarize, the additional trainable parameters introduced by deeper layers
> reduce the impact of misaligned initialization on the eigenvalues and enable the model to adapt more effectively to the
> structure of the true signal, thereby improving generalization performance.
>
> To provide a more detailed theoretical justification, let’s revisit the parameterization of our model: $\theta_j = a_j
> b_j^D \beta_j$ for $j \geq 1$, where the initialization is given by $a_j(0) = \lambda_j^{1/2}$, $b_j(0) = b_0$,
> and $\beta_j(0) = 0$. In this framework, the term $a_j b_j^D$ is responsible for learning the eigenvalues, while
> $\beta_j$ captures the signal. Then, Proposition 3.4 provides key insights:
>
> 1. **For signal components (large $\theta_j^\star$)**: $a_j b_j^D$ increases to approximate $|{\theta_j^*
>    }|^{\frac{D+1}{D+2}}$ multiplied by a constant factor. This indicates that the eigenvalues become better aligned
>    with the true signal as $D$ increases, since $\frac{D+1}{D+2}$ is an increasing function of $D$.
>
> 2. **For noise components**: If $\lambda_j \leq \epsilon^{2/(D+2)}$, $a_j b_j^D$ does not exceed its initial value
>    by more than a constant factor, thereby controlling the generalization error contributed by noise. A larger $D$
>    allows this condition to be met for more components, thus enhancing the model's generalization performance.
>
> While deeper layers facilitate more effective adaptation to the true signal, there are considerations regarding
> diminishing returns and optimal depth:
>
> 1. **Training Time and Computational Cost**: As shown in Corollary 3.3, the required training time $t \asymp
>    \epsilon^{-\frac{2D+2}{D+2}}$ increases with $D$, as does the computational cost due to the added layers. This may
>    not be desirable in practical applications where computational efficiency is a concern.
>
> 2. **Limited Benefit**: The primary benefit of deeper layers is to mitigate the impact of misaligned initialization on
>    the eigenvalues. However, once $D$ is sufficiently large and this impact is no longer dominant, further increasing
>    $D$ may not yield significant improvements.
>
> Therefore, the optimal depth is problem-dependent, with $D$ being large enough to minimize the influence of misaligned
> initialization while balancing computational considerations.

---

> > ### Comment · Reviewer_cuCj · 2024-08-13
> > **Re:**
> >
> > I thank the authors for responding to my comments. I'd like to increase my score based on the authors' response.

---

### Official Review · Reviewer_GugS · 2024-06-30

**Soundness:** 3
**Presentation:** 3
**Contribution:** 2
**Rating:** 6
**Confidence:** 3

**Summary:**

The authors analyze the generalization error for kernel regression sequence models in the overparameterized regime.
They rigorously show that due the overparameterization, the learned parameters are better adapted to the underlying structure.
They also provide some numerical experiments corroborating their findings (in the appendix).

**Strengths:**

- Clear motivation and presentation
- Rigorous analysis that could be useful for further research

**Weaknesses:**

The main concern here, is that the setting is too constrained, in that the kernel map is assumed to be fixed.

**Questions:**

- The NTK is mentioned multiple times in the introduction, could you elaborate a bit more on the connection of your work to NTK?
- How would you go about approximating the (infinite sequence ) $\lambda_i$ in practice? How does this affect performance and guarantees?

**Limitations:**

Limitations have been addressed.

---

> ### Author Rebuttal · Authors · 2024-08-06
>
> We appreciate your positive feedback and valuable comments.
> We will address your concerns in the following:
>
> > The main concern here, is that the setting is too constrained, in that the kernel map is assumed to be fixed.
>
> In fact, the goal of our paper is to investigate the **impact of the dynamic evolution of the kernel on the
> generalization performance**, so we are indeed considering a dynamic kernel/feature map.
> We have tried to consider a family of feature maps $\Phi_θ(\cdot)$
> parameterized by $θ$ and the model
>
> $$
> y=\Phi_θ(x)^{T}β+ε
> $$
>
> It is clear that it includes the neural networks as a sub-models and could (partially) capture the dynamic of feature
> learning characteristic in neural networks. It is clearly a hard task to investigate the most general adaptive
> kernel/feature models, however, some insight may be obtained from some simple families of feature maps.
>
> For the kernel regression, the feature map is fixed, namely, $\Phi_θ(x)= (\sqrt{λ_1}e_1(x)
> ,\sqrt{λ_2}e_2(x),...)$, where $(e_j)_{j\geq 1}$ are orthonormal functions, and the kernel is $K(x,y)=\Phi(x)^{T}\Phi(
> y)$.
>
> As the first attempt to understand the benefit of dynamic feature maps,
> we study the slightly complicated family $\Phi_θ(x)^{T}=(θ_1 e_{1}(x),θ_2e_{2}(x),....)$,
> where the parameters $θ$ are learned during the training process.
> However, there are some technical challenges to theoretically analyze the differential equation associated to the family
> $\Phi_θ(x)$. Thus, we simplify the analysis by transitioning to the sequence model, which is justified by the celebrated Le Cam equivalence.
>
> Here, we point out that while the eigenfunctions are fixed, **the feature map can change by learning the eigenvalues**,
> and, as shown in the main results, **this dynamic evolution of the eigenvalues can greatly improve the generalization**.
> Therefore, while simplified, our model still **capture the dynamic nature of kernels/feature maps** and
> provides insights and a solid theoretical foundation understanding more complicated over-parameterized models.
>
> From a higher perspective, our main insight is that *over-parameterized models combined with gradient-based training
> methods lead to a dynamically adaptive kernel that can adapt to the structure of the truth signal*,
> which we would like to refer to as the *"adaptive kernel" perspective*.
> Starting from this point,
> we believe we will be able to explore more realistic settings and more complex models in the near future.
>
> Moreover, as an extension of this work, with extra technicalities, we are now able to analyze directly the
> parameterization
> $\Phi_θ(x)^{T}=(θ_1 e_{1}(x),θ_2e_{2}(x),....)$ in the Reproducing Kernel Hilbert Spaces (RKHS) (see the Future Work
> section).
> Due to the paragraph limit, we plan to explore it further in an extended journal version of this work.
>
>
> #### Questions
>
> 1. > The NTK is mentioned multiple times in the introduction, could you elaborate a bit more on the connection of your
>    > work to NTK?
>
>    The motivation of this work is to go beyond the fixed kernel in the NTK theory and explore how the dynamic evolution
>    of the kernel improves the generalization performance.
>    The NTK theory shows that the training dynamics of deep neural networks can be approximated by certain kernel
>    regression with a fixed tangent kernel if the network width tends to infinity.
>    However, when the width is finite, the corresponding tangent kernel can be dynamically evolving during training,
>    which is shown empirically in recent works.
>    Therefore, in our work, we go beyond the NTK theory and explore the dynamic evolution of the kernel, particularly its
>    impact on the
>    generalization performance.
>    Since the analysis directly over fully-connected networks is extremely challenging, we simplify the setting by
>    considering parameterizing the kernel only by its eigenvalues and focusing on these eigenvalues.
>    Our results show that such dynamic evolution of the kernel can greatly improve the generalization performance
>    compared to the fixed kernel.
>
> 2. > How would you go about approximating the (infinite sequence $\lambda_i$) in practice? How does this affect
>    > performance and guarantees?
>
>    In our setting where we parameterize the kernel by the eigenvalues, we actually choose an infinite sequence
>    $\lambda_i$ first as the eigenvalues and then get the corresponding kernel, so the approximation of the eigenvalues
>    is not needed.
>    The usage of "eigenvalues" is just to be consistent with the fixed kernel framework (see the connection to NTK), and
>    these "eigenvalues" are just the initialization of the (dynamic) kernel in our setting.
>    Following our main result (Theorem 3.1 and Theorem 3.2), it suffices to choose eigenvalues with a fast decay such as
>    $\lambda_i = i^{-4}$ so that the extra error caused by the misalignment of the initial eigenvalues and the truth
>    signal is small.

---

> ### Comment · Reviewer_GugS · 2024-08-12
>
> I thank the authors for answering my questions. I keep my score.

---

### Official Review · Reviewer_kYgJ · 2024-07-12

**Soundness:** 3
**Presentation:** 3
**Contribution:** 4
**Rating:** 7
**Confidence:** 3

**Summary:**

This paper proposed an overparameterized gradient descent method. Its benefits on generalization has been verified both theoretically and empirically.

**Strengths:**

1.The paper is well-written with a clear structure. Motivations are well-explained on why the authors study the problem, and the illustrative examples are helpful in understanding the motivations.

2.Theoretical results are solid and well-organized. The authors made the theoretical settings clear: definitions are well-explained and assumptions are clear. Proofs of the theory are sound as far as I read into.

3.Experiments are provided in Appendix, which can verify the theoretical conclusions.

**Weaknesses:**

1.The insightful explanations for why the proposed method can improve generalization are lacked.

2.While the experiments in appendix have verified the effectiveness of overparameterized gradient descent, it is recommended to compare the performance of this new method with other algorithms, such as SGD or SAM.

**Questions:**

See weakness.

**Limitations:**

No negative social impact.

---

> ### Author Rebuttal · Authors · 2024-08-06
>
> Thank you for your positive feedback and recognizing the contribution of our work.
> We will address your concerns in the following:
>
> 1. > The insightful explanations for why the proposed method can improve generalization are lacked.
>
>    We are sorry for not presenting it clearer.
>    Generally speaking, the proposed over-parameterized gradient descent can improve generalization by adjusting the "
>    eigenvalues" to fit the structure of the truth signal, so that it can outperform the standard gradient descent.
>    In Section 3.3 paragraph "Learning the eigenvalues", we have further investigated why the over-parameterized gradient
>    descent can improve generalization. Proposition 3.4 in this paragraph shows that for the signal components, the
>    eigenvalues are learned to be at least a constant times a certain power of the truth signal magnitude;
>    while for noise components, the eigenvalues do not exceed the initial values by some constant factor. Therefore, the
>    over-parameterized gradient descent effectively adapts eigenvalues to the truth signal while mitigating overfitting
>    to noise, which leads to better generalization.
>
>
> 2. > While the experiments in appendix have verified the effectiveness of overparameterized gradient descent, it is
>    recommended to compare the performance of this new method with other algorithms, such as SGD or SAM.
>
>    Thank you for pointing out this.
>    We have conduct more experiments to compare the performance of the proposed over-parameterized gradient descent with
>    other algorithms, such as SGD or SAM.
>    The results show that without the over-parameterization, neither SGD nor SAM can achieve the optimal generalization
>    rate as the over-parameterized gradient descent does.
>
>    Theoretically, as far as we know, the SGD algorithm in kernel regression achieves the same generalization rate as the
>    standard gradient descent ([Lin and Volkan, 2020]), so that it also suffers from misalignment of the eigenvalues in
>    our setting and be inferior to the over-parameterized gradient descent.
>
> #### References
>
> [Lin and Volkan, 2020] Lin, Junhong, and Volkan Cevher. “Optimal Convergence for Distributed Learning with Stochastic
> Gradient Methods and Spectral Algorithms.” Journal of Machine Learning Research 21 (2020): 147–1.

---

### Official Review · Reviewer_EWHq · 2024-07-12

**Soundness:** 3
**Presentation:** 3
**Contribution:** 3
**Rating:** 6
**Confidence:** 3

**Summary:**

This paper investigates how over-parameterization can enhance generalization and adaptivity within the non-parametric regression framework. Drawing on insights from kernel regression and over-parameterization theory, the authors focus on the sequence model, which can approximate a wide range of non-parametric models, including kernel regression.

The authors highlight three key findings through mathematically stylized examples. First, they identify limitations of traditional kernel regression, including neural tangent kernel (NTK) theory, by demonstrating that the alignment (ordering) of eigenvalues can significantly impact generalization properties, even when eigenfunctions are correctly specified (Section 2). Second, by analyzing the gradient flow for a two-layer Hadamard neural network corresponding to the sequence model, they claim that this method can dynamically adjust eigenvalues to adapt to the underlying structure of the signal, achieving nearly the oracle rate with suitable regularization (Section 3.1). Finally, they show that adding depth can mitigate the impact of eigenvalue initialization, thereby improving generalization capability (Section 3.2).

**Strengths:**

This work offers a clear and effective account of the limitations of the traditional kernel regression framework (including NTK theory) and the advantages of over-parameterized gradient descent. The simple yet concrete examples effectively support the authors' claims, and the accompanying intuitions are sensible and easy to understand. Additionally, the two main theorems (Theorem 3.1 and Theorem 3.2) are presented concisely and effectively, and the discussion in Section 3.3 is particularly helpful for interpreting the results.

**Weaknesses:**

While this work makes significant theoretical contributions and provides a fresh perspective on the benefits of over-parameterization, there are areas where it could be improved:

**1. Narrow/Restrictive Settings:** The sequence model and the parameterization in Eq. (8) and Eq. (14), viz., the Hadamard neural network, might be too simplistic. I am curious if the insights in this paper can be extended to more realistic parameterizations, such as fully connected networks.

**2. Dynamically Evolving Kernels:** This paper addresses issues with misaligned eigenvalues and the benefits of over-parameterized gradient flow to mitigate them. However, during training, the kernel itself (and thus its eigenfunctions) may evolve, as empirically evidenced by [Seleznova and Kutyniok 2022] and [Wenger et al., 2023]. While the authors mention in Section 4 that this is left for future work, it would be helpful to understand if the eigenvalue alignment discussed in this paper can shed light on over-parameterization in practical settings beyond the stylized examples provided.

**Questions:**

1. The meaning of the sentence in lines 172-175 is unclear. Could the authors elaborate on this? Additionally, it would be helpful if the sequence model were described earlier to enhance its comprehension. Perhaps the authors could include a background section or rearrange the description of the sequence model in lines 183-186 to be presented upfront, with appropriate references.

2. Regarding the adaptive choice of the stopping time discussed in lines 279-288, I can roughly follow the authors' point. However, it appears that there is an unspecified constant, which is not known a priori, needed to choose the stopping time $t \asymp \epsilon^{\frac{2D+2}{D+2}}$, making it not practically feasible either.  Thus, I am wondering if the advantage is more aligned with universality, rather than practicality.  Could the authors clarify this?

3. Including the numerical experiments from Appendix A into the main text would be beneficial, if possible. Additionally, could the authors design a simple experiment to support the practical relevance of eigenvalue misalignment and the discussions in this paper? For instance, would it be possible to demonstrate that it is common in real-world datasets that kernels are well-identified but eigenvalues are easily misaligned?

Typos/minor suggestions:
- Line 89: add transpose to $X$
- LIne 149: defines -> defined
- Line 205: Inspired -> Inspired by
- Line 235: shows the generalization error -> presents an upper bound for the generalization error
- Line 260: have -> present?

**Limitations:**

This is primarily a theoretical work, and the authors discussed the potential limitations of the work and potential future research directions.

---

> ### Author Rebuttal · Authors · 2024-08-06
>
> Thank you for your efforts in reviewing our paper and recognizing the contribution of our work.
> We appreciate your constructive feedback and will address your concerns in the following:
>
> > Narrow/Restrictive Settings
> >
> > Dynamically Evolving Kernels
>
> We fully agree with you that the dynamic evolving kernel/feature would explain the superiority of the neural network and
> this is our start point of this manuscript. We have tried to consider a family of feature maps $\Phi_θ(\cdot)$
> parameterized by $θ$ and the model
> $$
> y=\Phi_θ(x)^{T}β+ε
> $$
> It is clear that it includes the neural networks as a sub-models and could (partially) capture the dynamic of feature
> learning characteristic in neural networks. It is clearly a hard task to investigate the most general adaptive
> kernel/feature models, however, some insight may be obtained from some simple families of feature maps.
>
> For the kernel regression, the feature map is fixed, namely, $\Phi_θ(x)= (\sqrt{λ_1}e_1(x)
> ,\sqrt{λ_2}e_2(x),...)$, where $(e_j)_{j\geq 1}$ are orthonormal functions, and the kernel is $K(x,y)=\Phi(x)^{T}\Phi(
> y)$.
>
> As the first attempt to understand the benefit of dynamic feature maps,
> we study the slightly complicated family $\Phi_θ(x)^{T}=(θ_1 e_{1}(x),θ_2e_{2}(x),....)$,
> where the parameters $θ$ are learned during the training process.
>
> However, there are some technical challenges to theoretically analyze the differential equation associated to the family
> $\Phi_θ(x)$. Thus, we looked for some alternative approaches.
> Fortunately, the celebrated Le Cam equivalence could allow us to simplify the setting to a sequence model.
> (Please see the answer of Question 1 for details)
>
> Here, we point out that while the eigenfunctions are fixed, **the feature map can change by learning the eigenvalues**,
> and, as shown in the main results, **this dynamic evolution of the eigenvalues can greatly improve the generalization**.
> Therefore, while simplified, our model still holds a connection to the dynamic evolving kernel/feature models and
> provides
> insights and a solid theoretical foundation understanding more complicated over-parameterized models.
>
> From a higher perspective, our main insight is that **over-parameterized models combined with gradient-based training
> methods lead to a dynamically adaptive kernel that can adapt to the structure of the truth signal**,
> which we would like to refer to as the *"adaptive kernel" perspective*.
> Starting from this point,
> we believe we will be able to explore more realistic settings and more complex models in the near future.
>
> Moreover, as an extension of this work, with extra technicalities, we are now able to analyze directly the
> parameterization
> $\Phi_θ(x)^{T}=(θ_1 e_{1}(x),θ_2e_{2}(x),....)$ in the Reproducing Kernel Hilbert Spaces (RKHS) (see the Future Work
> section).
> Due to the paragraph limit, we plan to explore it further in an extended journal version of this work.
>
> ### Questions
>
> 1. > The meaning of the sentence in lines 172-175 is unclear. Additionally, it would be helpful if the sequence model
>    > were described earlier to enhance its comprehension.
>
>    We apologize for the confusion. In these lines, we are illustrating that **kernel regression can be simplified to a
>    sequence model using the so-called Le Cam equivalence**.
>    In detail, the Le Cam equivalence states that the minimax risk of estimating a function in a reproducing kernel
>    Hilbert space (RKHS) is equivalent to the minimax risk of estimating a sequence of coefficients in a sequence model.
>    Informally, if we multiply $\Phi(x_i)$ on both sides of the equation $y_i=\Phi(x_i)^{T}β+ε_i$ and take the mean over
>    $i=1,...,n$, we get
>    $\frac{1}{n}\sum_i \Phi(x_i)y_i = \frac{1}{n}\sum_i \Phi(x_i)\Phi(x_i)^T β + \frac{1}{n}\sum_i \Phi(x_i)ε_i$.
>    Using the orthogonality of $e_j$'s in $\Phi(x)^{T}=(\sqrt{λ_1}e_1(x)   ,\sqrt{λ_2}e_2(x),...)$ and approximating the
>    empirical mean with the integration, we derive $\frac{1}{n}\sum_i \Phi(x_i)y_i \approx \Lambda β+\xi$, where $\xi$ is
>    a vector of Gaussian noise and $\Lambda$ is a diagonal matrix with entries $\lambda_j$, so we reach at the Gaussian
>    sequence model.
>    This equivalence allows us to focus on a more tractable model while retaining the essential characteristics needed
>    for our theoretical analysis. We hope this would clarify the point.
>
>    In the new revision, we will reorganize the content and make sure the model is properly introduced.
>
> 2. > Regarding the adaptive choice of the stopping time discussed in lines 279-288, I can roughly follow the authors'
>    > point.
>
>    We apologize for the confusion. Our point is to show the over-parameterized gradient descent method has also the
>    advantage of adaptivity over the vanilla gradient descent method.
>    For the vanilla gradient descent, the optimal stopping time $t \asymp ε^{-(2q\gamma)/(p+q)}$ is dependent on
>    the unknown parameters $p$ and $q$ of the signal; in contrast, for the over-parameterized gradient descent, the
>    choice $t \asymp ε^{-\frac{2D+2}{D+2}}$ does not require prior knowledge of the signal's structure. Moreover,
>    the hidden constant in this stopping time expression only need to be large enough with only a dependency on
>    Assumption 1, a weak assumption on the span of the signal (but not its specific structure).
>    So, **the stopping time is adaptive in the sense that it does not require prior knowledge of the signal's specific
>    structure**.
>
> 3. > Including the numerical experiments from Appendix A into the main text would be beneficial, if possible. ...
>
>     Thank you for your advice. With the additional content page if our paper is accepted, we will be able to include the
>    numerical experiments from Appendix A into the main text to enhance the readability of the paper.
>
>    For real-world datasets, we believe that the eigenvalue misalignment is a common phenomenon in practice.
>
>    *Due to the character limit, the details are given in a new comment block.*

---

> ### Author Response · Authors · 2024-08-07
> **Remaining response for Question 3**
>
> 3. > Including the numerical experiments from Appendix A into the main text would be beneficial, if possible. ...
>
>    Thank you for your advice. With the additional content page if our paper is accepted, we will be able to include the
>    numerical experiments from Appendix A into the main text to enhance the readability of the paper.
>
>    For real-world datasets, we believe that the eigenvalue misalignment is a common phenomenon in practice.
>
>     * From the theory side, as discussed in Example 2.2 (Low-dimensional example), if the regression function of the
>       data exhibits an unknown low dimensional structure on the covariates, then using a fixed kernel that takes all
>       covariates into account would lead to misalignment of the eigenvalues.
>       Since the low-dimensional structure is very common in real-world datasets, the eigenvalues would be misaligned in
>       practice.
>
>     * To conduct experiment on real-world datasets,
>       we can estimate the regression function's coefficients over the eigen-basis of the kernel and check whether they
>       exhibit a sparse structure.
>       We have conducted new experiments on datasets like "California Housing" and "Concrete Compressive Strength", where
>       we
>       choose the eigen-basis to be the multi-dimensional Fourier basis.
>       The results show that the coefficients over the eigen-basis concentrate over a few components, indicating the
>       sparse structure of the regression function,
>       so misalignment of the eigenvalues would occur.
>       Therefore, the eigenvalue misalignment might be very common in real-world datasets.

---

> > ### Comment · Reviewer_EWHq · 2024-08-11
> > **Response to the Authors' Rebuttal**
> >
> > I thank the authors for addressing my questions and concerns. Since most of my inquiries were for clarification, I trust that the authors will incorporate the additional explanations and reorganization into their revision. However, I was unable to locate the real-world dataset experiments mentioned in response to my Question 3;  if the experiments have already been conducted, could the authors attach them to the global rebuttal? It would be also beneficial to include these (along with the authors' response to my Q3) in the revision.
> >
> > Overall, I believe this work makes a solid theoretical contribution, though I suggest some revisions to further improve clarity regarding the scope of contributions and limitations, as well as their relation to existing work (e.g., following the authors' discussion with Reviewer EWHq and Reviewer kYgJ) and future directions. Specifically, I recommend the following among others, some of which the authors have already committed to:
> >
> >   (i) Properly introduce the sequence model, including a concise yet self-contained exposition of the Le Cam equivalence, potentially in the Appendix if necessary.
> >
> >   (ii) Further clarify the dynamic adaptivity of the kernel. While I agree with the authors that a dynamic update of eigenvalues can lead to adaptive kernel choices, even with a fixed eigenbasis, this offers limited adaptivity compared to a more general adaptive kernel/feature model, as the authors also noted. It would be helpful if the authors could elaborate on the extent to which the current approach is effective and whether they believe an extension to a more general kernel model is needed.
> >
> > With that understanding, I am inclined to raise my rating to a 6.

---

> ### Author Response · Authors · 2024-08-12
> **Experiment details**
>
> We would like to thank you for raising the score and providing further valuable comments on our paper.
> In the new revision, we will follow your advice to improve the clarity.
>
> For the experiments on the real-world dataset, it seems that we missed the opportunity to post a global rebuttal where we can show figures and tables.
> We apologize for this. We will include the results in the new revision of the paper.
> Nevertheless, we would like to describe in detail the experiments and results here.
>
> First, as discussed in Example 2.2 (Low-dimensional example) or Example 2.3, the misalignment happens when the order of the eigenvalues of the kernel mismatches the order of coefficients of the truth function.
> For the experiment setup, we consider the multidimensional Fourier basis (the trigonometric functions) as the eigen-basis of the kernel,
> which can be given by $e_{\mathbf{m}}(x) = \exp(2\pi i \langle{\mathbf{m},x}\rangle)$, where $\mathbf{m} \in \mathbb{Z}^d$.
> For the commonly considered kernel that correspond to Sobolev space (see also Example 2.2), the kernel and also its eigenvalues are isotropic in the sense that
>  $\lambda_{\mathbf{m}} \asymp (1+||{\mathbf{m}}||_2^2)^{-r}$ for some $r > 0$.
> Therefore, it gives an order of the eigen-basis.
>
> For the real-world dataset, we compute the coefficients of the regression function over the eigen-basis of the kernel by the empirical inner product.
> For "California Housing" and "Concrete Compressive Strength", the dimension of $x$ is 8, so we choose $\mathbf{m}$ up to $|m_i| \leq 2$, resuling in $5^8 = 390625$ coefficients.
> Then, we plot the magnitudes of the coefficients in the order given by the kernel.
> If the kernel is well aligned with the truth function, the coefficients should exhibit a smooth decay in this order.
> However, the resulting plot show **multiple significant spikes**.
> Also, among the coefficients, only very few components have large magnitudes, indicating the **sparse structure** of the regression function.
> Together, these results suggest that the eigenvalues of the kernel are misaligned with the truth function in these datasets.
>
> Please feel free to ask if you have any further questions or need more details on the experiments. Thank you.

---

### Official Review · Reviewer_QDTR · 2024-07-13

**Soundness:** 3
**Presentation:** 3
**Contribution:** 3
**Rating:** 6
**Confidence:** 3

**Summary:**

Authors consider the problem of fitting data with given a kernel but with a different estimator than kernel regression, which is based on an overparameterized version of the gradient flow for the squared loss. This estimator is inspired by the behavior of first-order algorithms in the area of deep learning theory under overparameterized models. The interesting regime in their theoretical result is when the order of the coefficients of the target function (that we hope to learn) in the eigenbasis of the kernel does not match the decay in the eigenvalues of the kernel itself; in this regime, while the conventional kernel regression estimator is suboptimal and can have undesirable dimension dependencies, they show that their estimator achieves a better rate, and intuitively is able to exploit the low dimensionality and adapt the eigenvalues of the kernel to the coefficients of the target function, a task which kernel regression with a fixed kernel is unable to do. Their analysis starts with using the Le cam theorem on the equivalence of kernel fitting and sequence models, then instead of running the gradient flow for the L2 loss of the sequence model only on the functions coefficients, they also train on some hyperparameters for the sequence model that resemble the kernel eigenvalues.

**Strengths:**

The theoretical result is interesting and connects to areas, first one in statistics about JS estimators and benefits of alternative estimators than fitting the data in the l-2 sense (which is equivalent to kernel regression in our context), and the deep learning theory literature where researchers are trying to figure our the benefits of training neural nets over fitting a fixed kernel, and what the role of overparameterization is in this regard.

Even though the algorithm is gradient flow and not computable in this paper, in a way it illustrate the benefit of training a simple overparameterized model and how it can is picking a kernel adaptively which works better than kernel regression with a fixed kernel. There are similar results in deep learning theory literature such as these:

[1] “Optimization and Adaptive generalization of Three layer Neural Networks”
[2] “What happens when SGD Reaches Zero Loss? – A Mathematical Framework”
[3] “Label noise (stochastic) gradient descent implicitly solves Lasso for quadratic parameterization”

**Weaknesses:**

My major concern is that similar results to this one already exist in the literature and the authors have not compared their results to them and how they differ from the them.In particular, running gradient-based methods on overparameterized nets, such as the quadratic parameterization is known to recover the underlying sparse solutions [3], also [2] section 6.2. In particular, I am curious if your result can imply or be phrased as a sparse recovery type argument for high high dimensional regression (i.e. similar rate to lasso)? And can you quantitatively compare your end results/rates with these papers, independent of the differences between algorithms/initialization? The work of [Zhao et al., 2022] that you have already cited also seems to have such sparse recovery results for high dimensional regression for gradient descent run on overparameterized model. Please also clarify the difference between your setup/rates with them quantitatively.

Minor comments
This work [1] also shows that training overparameterized three-layer neural nets with a specific architecture leads to the algorithm adaptively picking the kernel (instead of NTK where the kernel is fixed), which seems similar in flavor to your result, so it is good mention/compare.

**Questions:**

- Can you compare your rate with the results in sparse recovery, if you use lasso instead of your estimator? specifically do you think the low dimensional example that you mention can be observed as some kind of sparsity that can be exploited by lasso?

**Limitations:**

The literature review/comparison with current result still needs work, upon addressing this I am happy to increase my score.

---

> ### Author Rebuttal · Authors · 2024-08-06
>
> Thank you for your positive feedback and valuable comments.
> Following your advice, we will add the related papers that you mentioned.
>
> > My major concern is that similar results to this one already exist in the literature and the authors have not compared
> > their results to them and how they differ from them. I am curious if your result can imply or be phrased as a sparse
> > recovery type argument for high dimensional regression.
>
> Thank you for pointing out the related works and the connection between our result and sparse recovery.
> Indeed, it is known that the over-parameterized nets with gradient-based methods tend to recover the sparse solutions,
> as shown in the related works you mentioned and also [Hoff 2017], [Vaškevicius et al., 2019], etc., cited in our paper.
> However, there are major differences on the settings and the results between our work and the related works,
> which we will clarify in the following:
>
> * **Analysis of Generalization Error**:
>   The papers [2,3] and also previous works like [Hoff 2017], [Gunasekar et al. 2017] are focused only on the
>   optimization process using gradient-based methods, showing sparsity properties of the final solution,
>   but **they do not consider generalization error bounds**.
>   In comparison, our work (and also related works [Vaškevicius et al., 2019], [Zhao et al., 2022]) focuses further on
>   the generalization error of the over-parameterization methods, which turns out to be a harder problem.
>   More specifically, the works on the optimization process often assume the absence of noise in the data and consider
>   the final interpolator as $t \to \infty$.
>   However, such an interpolator may not generalize well under the presence of noise.
>   In fact, as shown in our work and also [Vaškevicius et al., 2019, Zhao et al., 2022], proper early stopping is also
>   crucial for generalization.
>   Therefore, we have to analyze the full training process rather than the final convergence point to understand the
>   generalization error, which is a more challenging problem.
>
> * **Linear regression vs. non-parametric regression** The most related works
>   are [Vaškevicius et al., 2019], [Zhao et al., 2022] that considered generalization performance of over-parameterized
>   models with gradient training under the setting of high dimensional linear regression.
>   In comparison, our work considers the non-parametric regression setting under reproducing kernel Hilbert spaces (
>   RKHS).
>   There are several main differences between our result and their results:
>     * *Problem settings*: They consider the high-dimensional linear regression where the input dimension (while growing)
>       is finite,
>       but we consider the non-parametric regression where the input dimension is directly infinite.
>       Also, they focus on the setting with the separation of signal components and noise components, while we consider
>       the more general setting of signal in the sequence model, allowing the magnitudes of the signal across components
>       to vary continuously.
>     * *Over-parameterization setup*:
>       [Zhao et al., 2022] considers the over-parameterization setup that $\theta = a \odot b$
>       where the initialization is $a(0) = \alpha \mathbf{1}$ and $b(0) = \mathbf{0}$;
>       [Vaškevicius et al., 2019] considers $\theta = u^{\odot D} - v^{\odot D}$ with $u(0) = v(0) = \alpha \mathbf{1}$.
>       However, their parameterization can not apply to the infinite dimensional case since the components are treated
>       equally and the $\ell^2$ norm of the estimator would be infinite after any step of training.
>       In comparison, our work considers parameterization $\theta_j = a_j b^D_j \beta_j$ with $a_j(0) = \lambda_j^{1/2}$,
>       $b_j(0) = b_0$ and $\beta_j(0)=0$,
>       so the components are treated differently.
>       Therefore, we have to tackle both the infinite dimensionality and also the interplay of different initialization,
>       which is more complicated.
>
>     * *Interpretation of the over-parameterization*: The previous works view the over-parameterization mainly as a
>       mechanism for implicit regularization,
>       while our work provides a novel perspective that over-parameterization adapts to the structure of the truth signal
>       by learning the eigenvalues.
>       Our perspective could provide more insights on the benefits of over-parameterization.
>
> In the new revision, we will add a detailed comparison and discussion with the related works in the paper.
> We hope this will clarify the novelty and contribution of our work.
>
>
> > Can you compare your rate with the results in sparse recovery, if you use lasso instead of your estimator?
> > specifically do you think the low dimensional example that you mention can be observed as some kind of sparsity that
> > can
> > be exploited by lasso?
>
> Our results can be phrased for the setting of high dimensional regression with sparsity.
> Taking a sparse signal $(\theta_j^*)_{j \geq 1}$ in our paper, e.g., $\theta_j^* = 1$ for $j \in S$, $|S| = s$ and
> $\theta_j^* = 0$ for $j \notin S$,
> we have $\Phi(\epsilon) = s$ and $\Psi(\epsilon) = 0$,
> so we can find that the resulting rate is $O(s/n)$, which is the same as the rate of the lasso estimator.
>
> For the low dimensional example in the sequence model, a JS estimator or thresholding estimator can indeed recover the
> sparse signal and achieve the optimal generalization rate, as shown in Chapter 6 and 8 in [Johnstone 2017]. We believe
> that a lasso-type estimator can also recover the sparse signal in this case. We think that a deeper connection between
> over-parameterization and these estimators can be explored in future work.
>
> ### References
> *Due to character limit, please see the next comment block for references.*

---

> > ### Comment · Reviewer_QDTR · 2024-08-14
> > **Response**
> >
> > Thank you for your response.
> >
> > regarding your claims in the new response that previous work in NN theory only show structural result about convergence of gradient based methods and does not consider generalization, that is not true I think, e.g. reference [1] that I sent above also consider the generalization capability of the final NN weights (which achieves similar rates to what you get in classical sparse recovery). So I think I don't understand that part of your claim.
> >
> > Regarding your comparison with Zhao et al., 2022, you mentioned they consider the l-2 case without weights so it cannot be used for the infinite weighted case, do you mean there is a fundamental barrier in generalizing their approach to cover the weighted infinite case, or is it that they just did not investigate this case in that paper?

---

> > > ### Author Response · Authors · 2024-08-14
> > >
> > > Thank you for your insightful comments.
> > >
> > > **Regarding the first point**, we apologize for any miscommunication.
> > > Our intention was to emphasize that while most research on over-parameterized models focuses on structural
> > > results, there has been less exploration into the generalization benefits of over-parameterization. We appreciate your
> > > reference to [1], which is indeed a significant work on the generalization capabilities of over-parameterized neural
> > > networks that we had missed.
> > >
> > > Our work shares some conceptual similarities with [1], particularly in viewing over-parameterization through an adaptive
> > > kernel perspective. However, there are still differences in our approach: while [1] explores an adaptive kernel space
> > > in the form of $G\odot K^\infty$ around the NTK space, our study examines an eigenvalue-parameterized kernel space with
> > > a fixed eigen-basis. We believe that the diversity of approaches, including those in [1], will contribute to a deeper
> > > understanding of the generalization properties in over-parameterized models. We will ensure to include [1] in our
> > > revised introduction and add a detailed comparison within the paper.
> > >
> > > **Regarding the second point**, there is indeed a fundamental difference between their approach and our approach in the weighted
> > > infinite case. Specifically:
> > >
> > > 1. We have to track the differently weighted initialization across components;
> > > 2. We provide new insights on the benefits of over-parameterization by learning the eigenvalues, see Proposition 3.4;
> > > 3. We extend our study to deeper layers, demonstrating additional benefits, whereas Zhao et al., 2022 is limited to the
> > >    two-layer setting.
> > >
> > > We appreciate your thorough review and the opportunity to improve our work.

---

> ### Author Response · Authors · 2024-08-07
> **References**
>
> [Hoff 2017] Peter D. Hoff. Lasso, fractional norm and structured sparse estimation using a Hadamard product
> parametrization. Computational Statistics & Data Analysis, 115:186–198, November 2017. ISSN 0167-9473. doi:
> 10.1016/j.csda.2017.06.007
>
> [Gunasekar et al. 2017] S. Gunasekar, B. Woodworth, S. Bhojanapalli, B. Neyshabur, and N. Srebro. Implicit
> regularization in matrix factorization. In Advances in Neural Information Processing Systems, volume 2017-December,
> pages 6152–6160, 2017.
>
> [Vaškevicius et al., 2019] Tomas Vaškeviˇcius, Varun Kanade, and Patrick Rebeschini. Implicit regularization for optimal
> sparse recovery, September 2019. URL http://arxiv.org/abs/1909.05122.
>
> [Zhao et al., 2022] Peng Zhao, Yun Yang, and Qiao-Chu He. High-dimensional linear regression via implicit
> regularization. Biometrika, 109(4):1033–1046, November 2022. ISSN 0006-3444, 1464-3510. doi: 10.1093/biomet/asac010.
>
> [Johnstone 2017] Iain M. Johnstone. Gaussian estimation: Sequence and wavelet models. 2017.

---

### Author Response · Authors · 2024-08-11
**Dear Reviewers**

We would like to thank the reviewers for their comments. We’ve taken the time to carefully go through each review. If anything is still unclear, please just let us know and we’ll get back to you quickly. Thanks again for your support!

---

### Comment · Area_Chair_RgZ8 · 2024-08-11
**Start discussions with authors, please**

Dear Reviewers,

Thank you for your valuable contributions to the review process. As we enter the discussion phase (August 7-13), I kindly request your active participation in addressing the authors' rebuttals and engaging in constructive dialogue.

Please:

- Carefully read the authors' global rebuttal and individual responses to each review.

- Respond to specific questions or points raised by the authors, especially those requiring further clarification from you.

- Engage in open discussions about the paper's strengths, weaknesses, and potential improvements.

- Be prompt in your responses to facilitate a meaningful exchange within the given timeframe.

- Maintain objectivity and professionalism in all communications.

If you have any concerns or need guidance during this process, please don't hesitate to reach out to me.

Your continued engagement is crucial for ensuring a fair and thorough evaluation process.
Thank you for your dedication to maintaining the high standards of NeurIPS.

Best regards,

Area Chair

---

### Decision · Program_Chairs · 2024-09-25

**Decision:**

Accept (poster)

**Comment:**

**Summary:**

This paper investigates how over-parameterization can enhance generalization and adaptivity in non-parametric regression models, particularly focusing on sequence models that approximate kernel regression. The authors introduce an over-parameterized gradient descent method to analyze how varying eigenfunction orders affects performance. Key findings include:

- Limitations of traditional kernel regression methods when eigenvalue alignment is suboptimal
- Over-parameterized gradient flow can adapt to the underlying signal structure and outperform standard methods
- Adding depth to the model further improves generalization by mitigating initialization effects

**Strengths:**

- Novel theoretical analysis providing insights into benefits of over-parameterization
- Clear motivation and presentation of ideas
- Rigorous mathematical framework that could be useful for further research
- Shows limitations of fixed kernel approaches like NTK and provides path forward
- Demonstrates how over-parameterization can lead to adaptive kernel learning

**Weaknesses:**

- Model setting is somewhat simplified/constrained compared to practical neural networks
- Limited experimental validation on real-world datasets and tasks
- Not fully clear how insights translate to more complex architectures used in practice
- Some reviewers felt theoretical claims about optimality of linear decay schedule needed stronger support

**After rebuttal:**

Overall, reviewers found the theoretical contributions and insights valuable, but had some concerns about how directly applicable the results are to practical deep learning. The authors provided reasonable responses explaining the scope of the work as a theoretical foundation to build upon. Most reviewers were satisfied with the responses and some increased their scores, indicating the paper makes a solid contribution to understanding over-parameterization, even if further work is needed to fully bridge to applied settings.